# Recurrent DNA break clusters drive replication-stress-induced copy number variants and genome diversification

Lorenzo Corazzi [1,2,8], Alex Ing[1,8], Eva Benito[3], Marco Raffaele Cosenza [3], Patrick Hasenfeld[3], Thomas Weber[4], Anna J. M. Marx[1,2], Vivien S. Ionasz [1], Nathan Trausch[1,2], Sarah Benedetto[5], Giulia Di Muzio[1,2], Boyu Ding [1,6], Jana Berlanda [1,2], Marco Giaisi[1], Nina Claudino[5], Thomas Höfer [5], Jan O. Korbel[3,7] & Pei-Chi Wei [1,2] ✉

Copy number variants (CNVs) are strongly implicated in neurological and psychiatric disorders and brain cancer, yet the process by which replication stress generates CNVs—and why some recur while others remain rare—remains poorly understood. Here, we show that recurrent DNA-break clusters (RDCs) act as common initiating lesions that drive both recurrent and non-recurrent CNVs. In murine neural progenitor cells subjected to chemically induced replication stress, bulk whole-genome sequencing identifies recurrent CNVs enriched at late-replicating RDCs within actively transcribed genes. Single-cell genome sequencing further uncovers frequent, non-recurrent CNVs associated with RDCs that arise during the transition from early to late DNA replication. These CNVs represent stable, heritable structural variants with breakpoints consistently enriched at RDCs. CRISPR/Cas9-mediated transcriptional suppression abolishes both RDC formation and CNV generation, establishing RDC-associated breaks as a shared upstream source. Mechanistically, CNV formation depends on DNA repair context: CNVs are Pol θ-dependent in NHEJ-deficient cells but arise independently of Pol θ in NHEJ-proficient cells. Together, these findings define RDCs as central drivers of replication-stress-induced genome diversification.

Copy number variants (CNVs) are a pervasive form of genetic structural variation involving deletions, insertions, or duplications of DNA segments[1]. Somatic CNVs can vary widely in size, from kilobases to entire chromosomal arms. CNVs represent a major source of genetic diversity, impacting both normal phenotypic diversity and the pathogenesis of various genetic disorders, including cancer[2–5]. Hence, unraveling the mechanisms behind CNV formation is crucial to obtain a full understanding of genetic variation and to improve diagnostic and therapeutic strategies for CNV-associated disorders. Multiple lines of evidence show that CNVs are dependent on DNA double-strand breaks (DSBs)[6,7]. In the context of pathology, replication stress plays a significant role in cancer, and certain neuropsychiatric disorders[8,9]. DNA replication stress often leads to stalled replication forks, which can subsequently collapse into DSBs. Although these DSBs have been

[1]Brain Mosaicism and Tumorigenesis Laboratory, German Cancer Research Center, Heidelberg, Germany. [2]Faculty of Bioscience, Ruprecht-Karl-University of Heidelberg, Heidelberg, Germany. [3]European Molecular Biology Laboratory (EMBL), Genome Biology Unit, Heidelberg, Germany. [4]Data Science Centre, European Molecular Biology Laboratory (EMBL), Heidelberg, Germany. [5]Division of Theoretical Systems Biology, German Cancer Research Center, Heidelberg, Germany. [6]Faculty of Medicine, Ruprecht-Karl-University of Heidelberg, Heidelberg, Germany. [7]Bridging Research Division on Mechanisms of Genomic Variation and Data Science, German Cancer Research Center, Heidelberg, Germany. [8]These authors contributed equally: Lorenzo Corazzi, Alex Ing. ✉e-mail: p.wei@dkfz-heidelberg.de

proposed as a source of CNVs[7], with distinct forms of replication stress producing divergent CNV patterns[10], the factors that distinguish DSBs leading to non-recurrent versus recurrent CNVs remain unclear. Further investigation into the precise interplay between DNA replication stress and DSB formation is therefore essential to fully understand the fundamental origins of CNVs, and the disorders they are responsible for.

A major debate is why CNV hotspots cluster at long, actively transcribed, late-replicating genes. DNA polymerase inhibition–induced replication stress preferentially generates CNVs at transcriptionally active loci, implicating transcription–replication collisions in CNV formation[11]. Transcriptional activity at specific genomic loci has even been shown to delay replication timing in vertebrate cell lines, including human cells[12]. These affected loci are typically late-replicating regions, leading to the hypothesis that delayed DNA replication may underlie CNV susceptibility[11,13]. However, it has also been shown that genome-wide transient inhibition of transcription activity does not alter replication timing in CNV-prone genomic regions[14], suggesting that while late replication timing and CNV formation often coincide, they are not necessarily causally linked. The precise contribution of transcriptional activity, replication timing, and their interplay in the formation of CNVs remains to be elucidated.

In mammalian cells, a substantial number of long genes are hotspots for recurrent DNA-break clusters (RDCs)[15–17]. The majority of these long genes encode proteins involved in neural adhesion and synaptic plasticity, suggesting that RDCs may play a role in regulating neurogenesis[18]. RDCs exhibit hallmarks of transcription–replication conflicts, as they consistently occur within actively transcribed genes. These breaks arise where transcription and replication collide: DSBs accumulate in regions where the two machines move in opposite directions. The DSB orientation tracks the direction of replication forks, implying formation at stressed forks. When forks converge (inward-moving forks), the breaks align head-to-head, a 3-D configuration typically resolved by homologous recombination or end-joining[19]. We hypothesize that this configuration promotes deletions and copy number loss in the RDC-containing genes. Consistent with this idea, around one-third of RDC-containing genomic loci were

hotspots for CNVs identified in neurons[15,18]. We hypothesize that densely distributed breaks within RDCs generate multiple breakpoint combinations, producing recurrent, intra-RDC CNVs and rare, non-recurrent CNVs between RDCs genome-wide.

In this work, we characterised RDCs as a major driver of CNV formation and genome diversification under conditions of replication stress. By combining deep whole-genome sequencing, single-cell genomics, DNA-break mapping, and CRISPR/Cas9-mediated transcriptional manipulation, we found that both recurrent and rare CNVs have their origins in RDC regions. We also identified a context-dependent role for DNA polymerase theta (Pol θ) in shaping CNV formation, linking DNA repair pathway choice to somatic genome diversification.

## Results

### A subset of RDC genes are recurrent CNV hotspots

Our initial goal was to determine whether replication stress causes DNA copy-number changes at recurrent DNA-break clusters (RDCs). We treated mouse embryonic stem (ES) cell-derived neural progenitor cells with aphidicolin (Aph), a DNA polymerase inhibitor commonly used for inducing CNVs[11] and RDCs[18], with dimethyl sulfoxide (DMSO) used as a control[18]. In order to determine the effect this condition has on DSB and CNV formation (Fig. 1A), we carried out whole genome sequencing (WGS). In the following experiments, unless otherwise indicated, the neural progenitor cells were deficient in Xrcc4 and p53, a genotype combination that enhances DSB recovery, as described in our previous investigations[16,20]. We performed WGS at an average 180-fold coverage of neural progenitor cell genomes to capture heterogeneous CNV patterns. The genomes of neural progenitor cells were predominantly diploid, with the exception of a chromosome 8 gain that was independent of aphidicolin treatment (Supplementary Fig. 1). Comparative read-depth analysis revealed a consistent, genome-wide under-representation of RDC loci in Aph-treated cells. When the log₂(Aph/DMSO) average ratios for all 152 RDCs were rank-ordered, five loci fell at least three standard deviations below a size-matched random distribution (Fig. 1B, C). In contrast, there were no increases of even two standard deviations, indicating that aphidicolin leads to

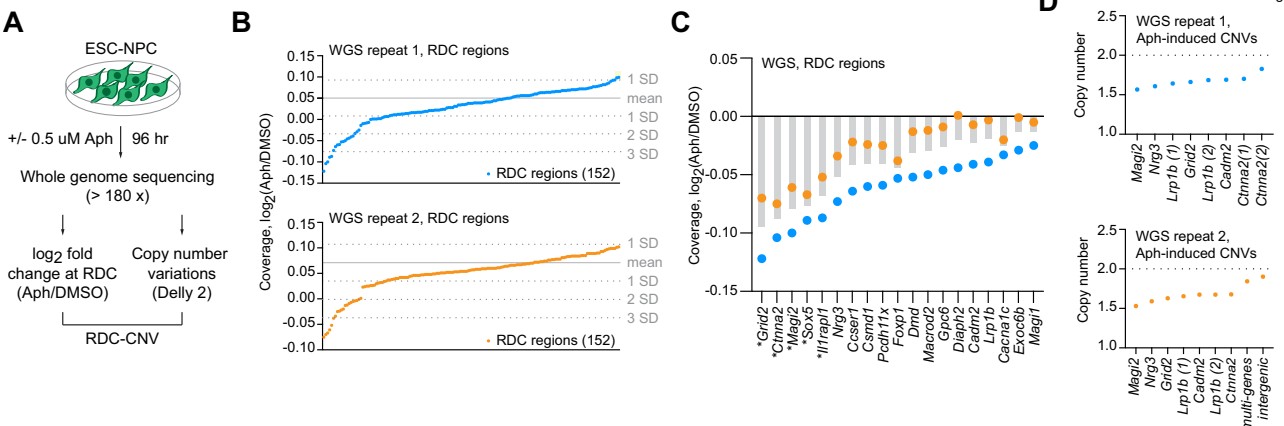

**Fig. 1 | Low-dose aphidicolin elicits shallow copy-number erosion at recurrent DNA-break clusters (RDCs) and discrete heterozygous deletions in ES cell-derived neural progenitors (NPCs). A** Experimental design. Mouse ESC-derived NPCs were treated with 0.5 μM aphidicolin (Aph) for 96 h, or with solvent control (DMSO) for the same period. Genomic DNA from two independent biological replicates was subjected to >180 × whole-genome sequencing (WGS). Read depth was analysed (i) for log₂(APH/DMSO) fold-change across 152 previously defined RDC intervals and (ii) for structural copy-number variants (CNVs) using Delly2; together these outputs are referred to as the RDC-CNV dataset. **B** Rank-ordered log₂(Aph/DMSO) coverage for every RDC in WGS replicate 1 (upper) and replicate 2 (lower). Colored dots, individual RDCs; grey solid line, mean of 1520 size-matched random regions excluding RDC regions; grey dotted lines, ±1–3 s.d. of the random set. **C** Per-gene view of the 19 RDCs showing the largest coverage losses (left to right; gene symbols on x-axis) in both replicates. Genes that deviate by more than 3 SD in Fig. 1 are indicated with a star. Grey bars show the average of replicate 1 (light blue dots) and replicate 2 (orange dots) values. **D** Copy number for CNVs detected in replicate 1 (top) and replicate 2 (bottom) by Delly 2. Each dot represents a discrete CNV; gene names denote overlap with an RDC from (**C**). The dashed line marks the diploid baseline (2 copies); all events correspond to single-copy losses (median ≈ 1.6 copies). Source data are provided as a Source Data file.

losses rather than amplifications at these sites, in line with previous findings[11]. Under a null model in which CNV formation is independent of RDC location, the probability that all six recurrent CNVs would lie within RDCs by chance is $p = 5.33 \times 10^{-15}$ (hypergeometric test; population ≈35,771 genes; successes = 152). Thus, while recurrent CNV hotspots are rare at a genome-wide scale, they are strongly enriched at RDC-containing, ultra-long, late-replicating neuronal genes, reinforcing the conclusion that transcription-dependent RDCs provide a proximate source of CNV formation under aphidicolin-induced replication stress. Closer inspection of the most strongly affected genes showed concordance between replicates (Fig. 1C). When genes were paired between replicates, correlation in loss of read-depth coverage was very high (Pearson's $r = 0.93$, $p = 1 \times 10^{-6}$ permutation tested). The top tier of depleted loci is dominated by very large, neuronally expressed genes such as *Grid2*, *Ctnna2*, *Magi2*, *Ilrapl1* and *Sox5*, which are classical hotspots for replication-associated fragility[11,18].

Structural-variant calling corroborated these findings. Across both datasets, we identified nine discrete CNVs using the Delly2[21] variant calling tool (Fig. 1D and Supplementary Data 1). All CNVs exhibited relative copy numbers of 1.5 - 1.8, consistent with subclonal deletions. Each of these six deletions mapped individually to a distinct RDC-containing gene, showing significant DNA sequence loss (*Magi2*, *Nrg3*, *Lrp1b*, *Grid2*, *Cadm2*, and *Ctnna2*). Two CNVs mapped in replicate 2 (one multigenic and one intergenic) were not reproducible and were therefore excluded from the following analysis. Together, these data show that low-dose aphidicolin produces two hierarchies of instability at RDC genes: (i) a pervasive, shallow depletion of sequencing coverage across roughly one-fifth of all RDCs, suggestive of incomplete genome duplication or sub-clonal loss, and (ii) rarer, focal deletions confined to a subset of the most vulnerable loci. The concordance between biological replicates underscores the reproducibility of these events.

### Heterogeneous DNA sequence loss within RDC-CNV hotspots

Neural progenitor cells exposed to aphidicolin revealed discrete copy-number losses in six RDC-containing loci, situated within late Constant-Timing Regions (CTRs), where multiple replication units simultaneously complete DNA replication within a single region (Fig. 2A). We noted that Delly2 has a cutoff that could not recover all genomic areas showing copy number loss. In some cases, the maximum loss in read depth reaches almost the entire allele (*Ilrapl1*, *Ccser1*; Supplementary Fig. 2A). At each locus, the log₂(Aph/DMSO) read-depth ratio dipped by ≈0.3–0.5, consistent with heterozygous deletions spanning 0.3–2 Mb. To precisely map DSBs in genes with CNVs, we re-plotted the RDC DSBs determined by capture-ligation based assay, linear amplification-mediated, high-throughput genome-wide translocation sequencing, LAM-HTGTS[16,22]. This assay mapped unidirectional clusters of single-ended DSBs precisely within these deleted intervals[16], indicating that aphidicolin-induced fork stalling converts recurrent breakage into segmental loss. Despite the structural change, high-resolution Repli-seq profiles showed only a minor shift in replication timing: the affected regions remained uniformly late-replicating under both solvent and stress conditions. In contrast, we did not detect CNVs at RDC regions located within replication timing transition regions (TTRs; Supplementary Fig. 2B) or biphasically replicating domains (Supplementary Fig. 2C), the latter of which have been associated with common fragile site formation[23].

Not all RDC loci in the late CTRs followed this trajectory. Three other late CTR loci (*Adgrl3*, *Pcdh9*, and *Csmd1*) accumulated equally robust break clusters after aphidicolin yet did not suffer significant copy number loss (Fig. 2B). To test if this difference was caused by under-replication, we calculated under-replication indexes (URI)[14] for RDCs located in the late CTR. Comparing Aph-treated to solvent-treated (DMSO) cells, we found that not all CNVs correlated with under-replication, defined by low URI scores (Supplementary Fig. 2D).

In addition, we also observed cases where low URI at the *Csmd1* and *Nkain2* loci did not account for CNV formation, and vice versa (Supplementary Fig. 2D), suggesting that these differences were not solely due to incomplete DNA replication.

These observations indicate that replication-stress-induced breaks at fragile sites are not entirely replication timing-dependent, and can be resolved through locus-specific pathways.

### Large single-cell structural variants co-localize with RDCs

Bulk analyses revealed heterogeneous copy number losses at RDC-linked loci, suggesting that recurrent DNA-break clusters act as initiating lesions that diversify, rather than merely mark, genome instability. We therefore hypothesized that replication-associated breaks at RDCs are resolved into allelic deletion-type structural variants that segregate into daughter cells. We used Strand-seq, a single-cell method that preserves template-strand inheritance, to directly detect these deletion-associated variants[24–26].

We generated 79 high-quality single-cell libraries from untreated cells and 88 from aphidicolin-treated cells. Aphidicolin-treated cells exhibited a similar sister chromatid exchange (SCE) frequency (Aph vs. DMSO = 9.5 vs. 8.1 SCE per cell), indicating replication stress in the prior cell cycle did not affect global SCE levels. Next, we investigated whether the six CNV hotspots identified by bulk sequencing were preserved in daughter cells (Fig. 3B and Supplementary Fig. 3A). Aphidicolin-treated cells exhibited significant deletions at three of six CNV hotspots (Fig. 3B, C). Multiple independent cells exhibited distinct, single-allelic loss events at these loci, most prominently at *Lrp1b*. These deletions were significantly enriched compared to background (permutation test based on the Welch two-sample *t*-statistic, $p = 1 \times 10^{-4}$), confirming their recurrent nature. In contrast, no focal deletions at *Ctnna2* were detected, despite this locus being the most prominent CNV in bulk analyses, suggesting either limited sampling or negative selection against structural disruption of the locus.

To assess whether RDC-associated breaks coincide with non-recurrent copy number alterations in single cells, we performed an unbiased CNV analysis on cells with high-quality Watson/Crick strand signals. Cells previously exposed to aphidicolin frequently exhibited substantial DNA loss across the genome (Fig. 3D, E and Supplementary Fig. 3B), consistent with large, multi-megabase deletions. We therefore focused on cells with pronounced copy-number loss and mapped their large-scale genomic alterations. In 29 cells with reduced genome-wide copy number (mean < 1.92), 302 breakpoints were identified (Supplementary Data 2). A significant fraction of these were mapped to RDC regions (118/302, permutation test, $p = 1 \times 10^{-4}$). Notably, nearly half of these RDC-associated breakpoints overlapped with replication timing transition regions (TTRs) (51/118; permutation test $p = 6 \times 10^{-4}$), indicating that TTR-associated RDCs contribute to severe copy-number variants.

We also captured two cells that likely came from the same parental cell (Cell 9 and 10), as we observed reciprocal sister chromatid exchange events on chromosome 12 (Supplementary Fig. 3C). In one of the suspected daughters (Cell 9), we found only a short Watson peak on chromosome 2 while the Crick strand displayed significant copy number loss downstream of the Watson peak (Fig. 3F). This feature, as previously seen in hematopoietic stem cells and epithelial cell lines[27], represented a fold-back chromosome undergoing a break-fusion-bridge cycle. This event was not present on the other daughter cell (Cell 10), suggesting that parental cell exposure to replication stress can lead to the diversification of daughter cell genomes (Fig. 3G).

Taken together, these analyses demonstrate that RDC-associated breaks are not restricted to producing focal intragenic deletions, but can also precipitate complex and larger structural variants that extend across megabase-scale chromosomal regions. Rather than being fully repaired or averaged out in a population, lesions initiated at recurrent break clusters are converted into a spectrum of structural outcomes,

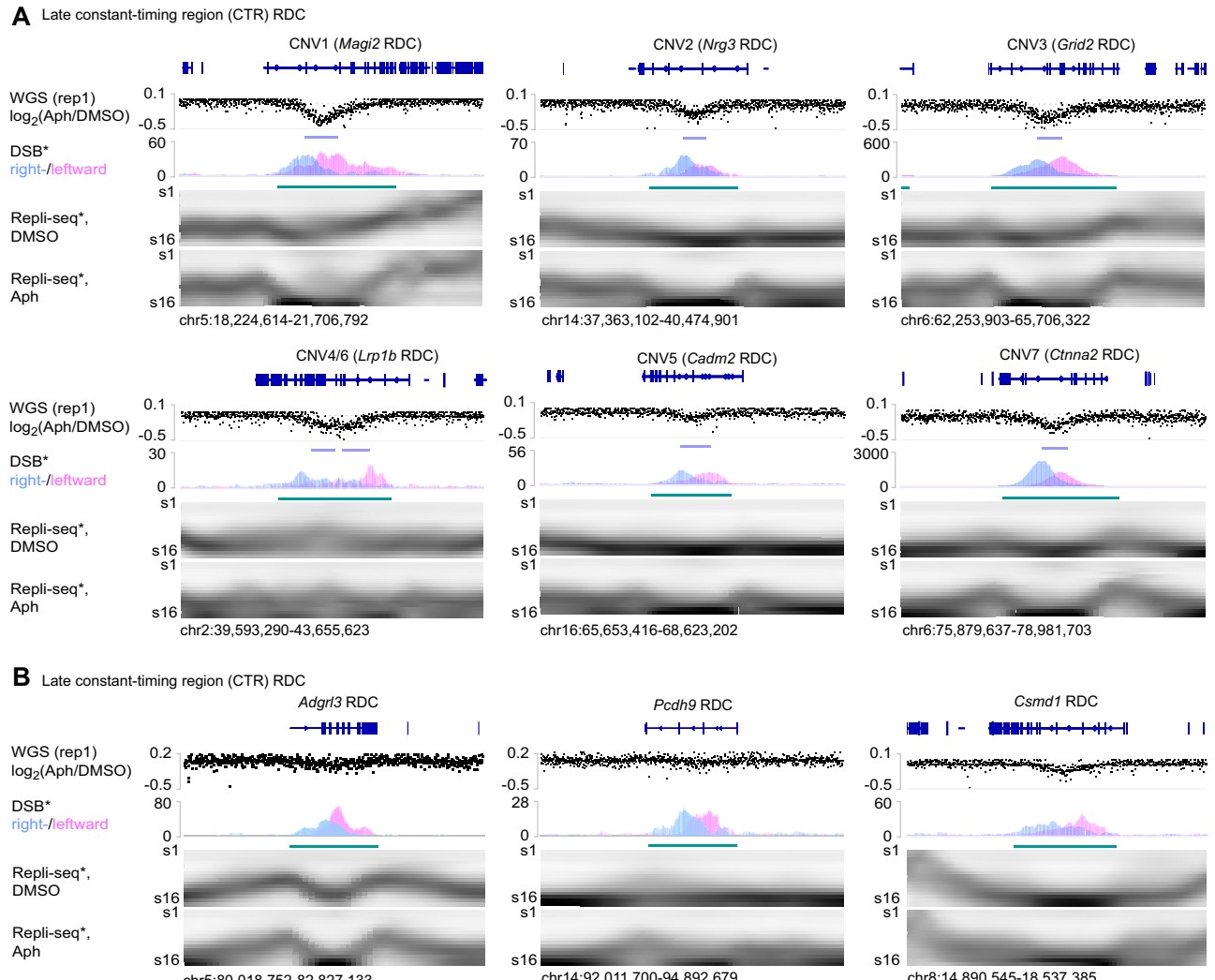

**Fig. 2 | Aphidicolin-induced heterozygous deletions arise at a subset of late-replicating constant-timing recurrent DNA-break clusters. A** Six copy-number variants (CNV1–CNV7) identified in neural progenitor cells after 96 h of aphidicolin (Aph) treatment. For each locus (gene names indicated above), the upper track shows the whole-genome-sequencing log$_2$(Aph/DMSO) coverage ratio (WGS, replicate 1); downward deflections mark heterozygous losses (turquoise bars). The middle track plots strand-specific LAM-HTGTS signal for single-ended double-strand breaks (DSB), coloured by right- (blue) and left-ward (pink) polarity; shaded distributions locate clustered breaks that coincide with CNV boundaries. The lower two heat maps present Repli-Seq replication-timing profiles for solvent control (DMSO) and Aph-treated cells (S-phase fractions S1–S16), illustrating that each CNV lies within a late constant-timing region (CTR), where replication timing remains unchanged upon Aph treatment. Data was extracted from a previous study[16] and re-plotted. Genomic coordinates are given beneath each panel. **B** Three additional late CTR RDC hotspots (*Adgrl3, Pcdh9, Csmd1*) accumulate Aph-induced DSB clusters below Delly2 significance cutoff. *Adgrl3* and *Pcdh9* did not show significant DNA copy number loss, while *Csmd1* showed significant DNA sequence losses (Fig. 1C). Tracks are as in (A).

from small intragenic deletions to arm-scale losses, which segregate into daughter cells. Thus, replication stress at RDCs acts as a driver of genome diversification, generating both locus-specific and large-scale structural variants that permanently remodel the genomes of neural progenitor cells.

### RDC-dependent CNVs occur without replication timing shifts

To test whether transcription of large neuronal genes is required for the formation of RDCs, we used CRISPR–Cas9 to delete the proximal promoter/enhancer of *Ctnna2* and *Nrg3* genes (Fig. 4A). Two independent neural progenitor cell clones derived from ES cell lines (pe-del1 - 4; Fig. 4B) were used for each gene locus, and the unique deletions were validated by Sanger sequencing (Supplementary Fig. 4). By mapping nascent RNA with the global run-on sequencing (GRO-seq) assay[28], we found that the nascent transcripts naturally encoded at the minus strand in the parental cells disappeared in the promoter/enhancer-deleted ES cell-derived neural progenitor cells (Fig. 4B, green

panels). Mapping DSBs in the same cell in parallel experiments showed that RDC clusters vanished from the *Ctnna2* and *Nrg3* loci in promoter/enhancer-deleted ES cell-derived neural progenitor cells (Fig. 4B, DSB panels). This data strongly indicates that RDCs at *Ctnna2* and *Nrg3* are transcription-dependent.

To test whether DNA sequence losses at CNVs were RDC dependent, we performed high-coverage (average 180×) whole-genome sequencing on neural progenitor cells with promoter/enhancer deletions, both before and after mild aphidicolin treatment (Fig. 4C). We observed consistent CNV losses at *Magi2, Ccser1, Grid2, Sox5, Gpc6, Prkg1*, and *Lrp1b* in the deleted clones (Supplementary Data 1). Significantly, the log$_2$(Aph/DMSO) ratio indicated that copy number losses at the *Ctnna2* (pe-del c1) or the *Nrg3* (pe-del c3) loci were no longer present in these clones (Supplementary Data 1). These data show that Aph-induced CNVs at the *Ctnna2* and *Nrg3* genes are dependent on their transcription, further suggesting that these CNVs result from RDCs.

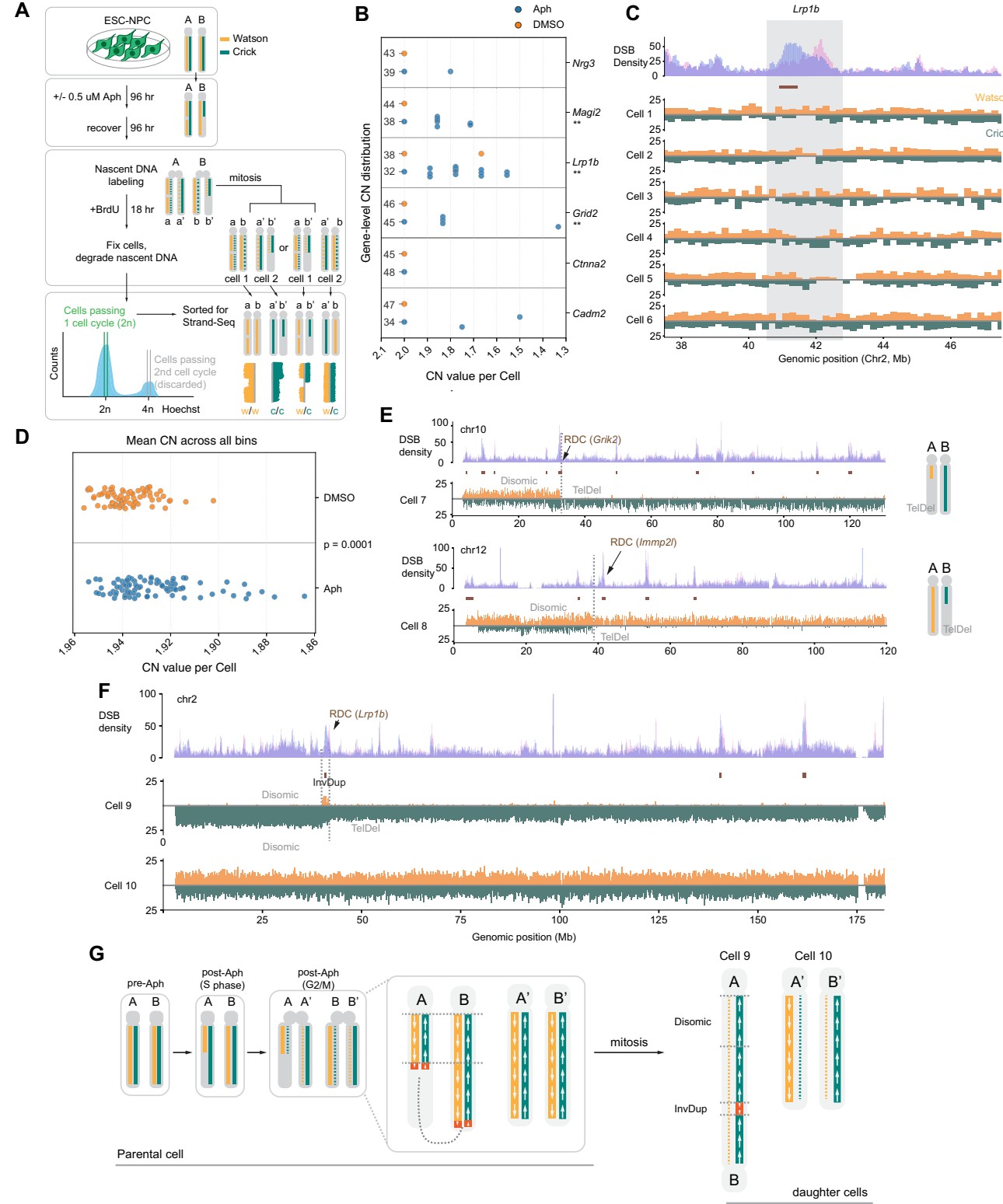

As shown in previous studies[12], transcription activity can delay replication, resulting in under-replicated genomes that coincide with CNV hotspots. In order to evaluate changes in DNA replication timing, we performed two-fraction replication sequencing (Repli-Seq) on the parental neural progenitor cells, as well as on the promoter/enhancer-deleted cell lines (Fig. 4D). At the *Nrg3* locus, transcription ablation did not further alter the replication timing. At the *Ctnna2* locus, the replication of the locus and the adjacent area was even delayed. These data indicated that the loss of

CNV from these cells was not the result of advanced replication timing.

Together, our data suggested that RDCs at the *Ctnna2* and *Nrg3* loci are transcription-dependent, and DNA sequence loss in CNV hotspots within the two tested gene loci is RDC-dependent.

**Pol θ promotes CNV in Xrcc4/p53-deficient neural progenitors**

Next, we sought to determine whether polymerase theta (Pol θ) activity regulates CNV formation in an NHEJ-deficient context. To this

**Fig. 3 | Replication Stress Induces Complex Structural Variants and Asymmetric Segregation at Recurrent Deletion Clusters. A** Experimental workflow for single-cell Strand-seq. Neural progenitor cells (NPCs) were treated with aphidicolin (Aph) for 96 h, allowed to recover, and labeled with BrdU for one DNA replication cycle prior to Strand-seq procedures to distinguish parental Watson and Crick template strands. **B** Dot plot showing gene-level copy number (CN) values for six RDC-associated genes in individual cells treated with Aph (blue) or solvent control (DMSO, orange). Aph-treated daughter cells frequently show CN losses at RDC loci, including *Lrp1b* (14 cells), *Grid2* (4 cells), *Magi2* (7 cells), *Cadm2* (2 cells), and *Nrg3* (1 cell). Statistical significance was assessed using a permutation test (10,000 permutations; one-sided, APH < DMSO) based on the Welch two-sample *t*-statistic, with cell labels shuffled between conditions. ** $P < 0.05$ (*Grid2* $P = 0.017$, *Lrp1b* $P = 0.0014$, *Magi2*, $P = 0.0076$). Source data are provided as a Source Data file. **C** Representative Strand-seq coverage tracks at the *Lrp1b* locus (chr2) in Aph-treated cells. Top track shows bulk DSB density. Shaded region indicates the *Lrp1b* gene body, displaying focal, haplotype-specific deletions (drops in read coverage on either Watson or Crick strands) that align with DSB clusters (a brown bar within

the area). **D** Scatter plot of mean genome-wide CN in different conditions. CN Aph-treated (blue) or control (DMSO, orange) individual cells are shown. *P* values were computed using a label-permutation test (10,000 permutations; one-sided, APH < DMSO, as described for (**B**), $P = 0.0001$. Source data are provided as a Source Data file. **E** Strand-seq examples of large-scale chromosomal terminal deletions (TelDel) whose breakpoints directly coincide with RDC sites (marked by DSB density peaks and brown horizontal bars aligned below the peaks). Examples shown for *Grik2* (chr10) and *Immp2l* (chr12). Cartoons on the right depict the resulting chromosomal structures. **F** Strand-seq analysis of two sister cells originating from the same parental cell (Cell 9 and Cell 10). Cell 9 displays a complex fold-back inversion-duplication (InvDup) and terminal deletion on chromosome 2, initiated at the *Lrp1b* RDC, while its sister cell is disomic at this locus. **G** Proposed model illustrating how replication-associated breaks at RDCs in parental cells can be resolved into asymmetric, complex structural variants, such as the InvDup/TelDel seen in (**F**) that segregate differentially into daughter cells following mitosis. Source data are provided as a Source Data file.

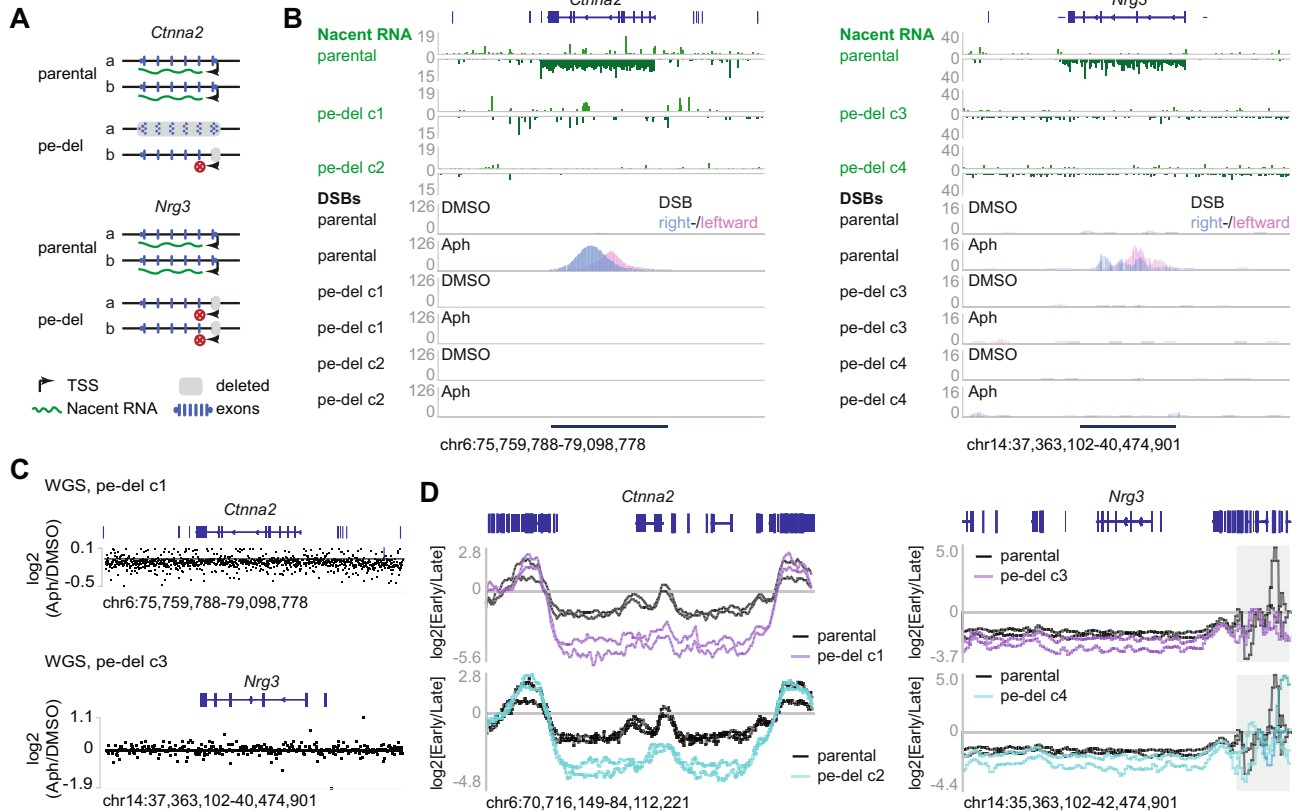

**Fig. 4 | Promoter/enhancer deletions extinguish transcription-dependent RDCs at the CNV-containing genes Ctnna2 and Nrg3 without elevating their replication timing. A** Strategy for CRISPR/Cas9 deletion of the major promoter/enhancer (pe-del) that drives the long neuronal isoform of the giant genes *Ctnna2* (left) and *Nrg3* (right). Grey boxes mark the region removed in two independent clones (c1, c2); red X markers indicate transcription inactivation. Isoform-specific transcription start sites (TSS) and exons are depicted below. **B** Loss of nascent transcription eliminates RDCs. Data (green) shows gene-body-spanning transcription of *Ctnna2* (left) and *Nrg3* (right) in parental cells, which is almost completely abolished in both pe-del clones. LAM-HTGTS tracks (blue/pink, right/left-

ward single-ended DSBs) reveal a sharp, aphidicolin-induced DSB peak in parental cells that disappears in clones with deleted promoters. Scale bars: DSB per thousand. **C** WGS log₂(Aph/DMSO) coverage ratio around the *Ctnna2* locus in the *Ctnna2* promoter/enhancer-deleted cells (pe-del c1, upper), and around the *Nrg3* locus in the *Nrg3* promoter/enhancer-deleted neural progenitor cells (pe-del c3). **D** Replication-timing (Repli-seq) profiles across the two loci. Log₂(Early/Late) ratios are plotted for parental cells (black) and the two pe-del clones in untreated conditions. The grey box at the *Nrg3* adjacent region denotes a repetitive sequence-rich area.

end, we treated Xrcc4/p53-deficient neural progenitor cells with aphidicolin alone, or with aphidicolin plus ART558[29,30], a small-molecule inhibitor of Pol θ DNA synthesis activity. Combined treatment with aphidicolin and ART558 abolished focal, recurrent CNV formation at six aphidicolin-dependent CNV loci (Fig. 5), suggesting aphidicolin-mediated CNVs were Pol θ-dependent in Xrcc4/p53-deficient cells. Unbiased, genome-wide copy-number profiling

showed that a subclonal chromosome 8 gain detected in untreated cells was lost, leading to reduced log₂ coverage ratios (Supplementary Data 1 and Supplementary Fig. 5A). We also detected a reproducible subclonal loss of chromosome 6, consistent with selection in culture. Further, we identified 46 additional reproducible CNVs (Supplementary Data 1). These CNVs were absent in cells treated with aphidicolin alone but were consistently detected in cells treated with

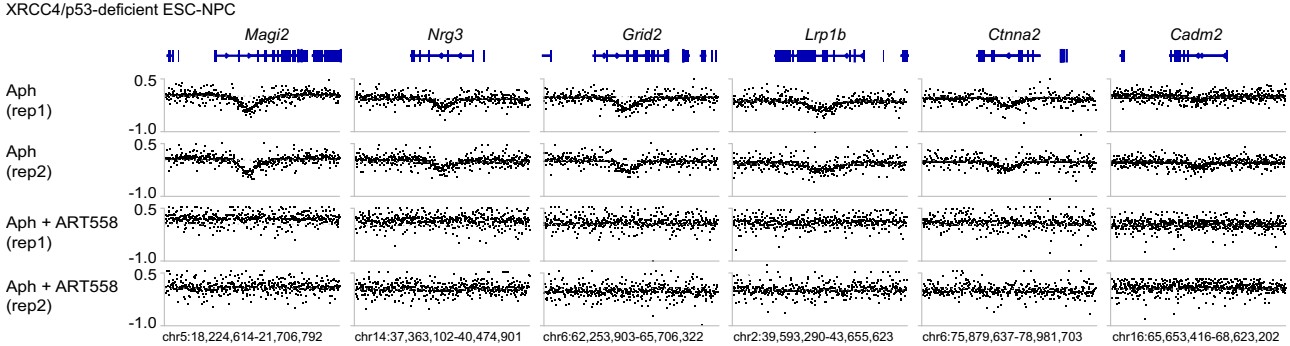

**Fig. 5 | Recurrent CNVs in Xrcc4/p53-deficient neural progenitor cells are Pol θ-dependent.** Whole-genome sequencing coverage ratios at six recurrent CNV loci identified in neural progenitor cells following 96 h of the indicated treatments. For each locus, binned sequencing coverage under each condition was normalized to the untreated control (DMSO) and plotted as log$_2$ ratios (y-axis). Independent experimental replicates are shown. Genomic coordinates and gene annotations are indicated for each track. Note that the Aph tracks were replotted from data presented in Fig. 2.

ART558, indicating that they arise specifically upon Pol θ inhibition. In addition, about half (20/46) of the Aph/ART558-induced CNVs were not in genes, suggesting they arise from an RDC-independent mechanism.

Because Pol θ is known to fill nascent DNA gaps at stalled replication forks, particularly on the lagging strand[31,32], we hypothesized that these copy-number alterations reflect differences in DNA content resulting from replication stalling. To test this, we examined replication timing at recurrent CNV loci in cells treated with both aphidicolin and ART558. We observed a significant correlation in which CNV gains were associated with early-replicating regions (Kendall's rank correlation, tau = 0.31, p = 0.002), whereas CNV losses were enriched in late-replicating regions, indicating that copy-number changes mirror replication timing (Supplementary Fig. 5B). This relationship was enhanced in cells treated with ART558 alone (Kendall's rank correlation, tau = 0.51, p < 2.2 × 10$^{-16}$; Supplementary Fig. 5C), supporting the notion that ART558-induced CNVs resulted from replication stalling.

## Pol θ promotes RDC-mediated translocation in Xrcc4/p53-deficient cells

It has been suggested that, in the non-homologous end-joining (NHEJ) deficient cells, DNA repair and chromosomal translocations are mediated by Pol θ[31–34]. To test whether Pol θ ligates replication stress-mediated DSBs at RDCs, we employed Linear amplification-mediated, high-throughput genome-wide translocation sequencing (LAM-HTGTS)[16,22] to map DSB junctions across the genome (Fig. 6A). Briefly, we introduced a CRISPR-Cas9 bait DSB on chromosome 6, 8, or 12 in Xrcc4/p53-deficient neural progenitor cells to capture spatially proximal prey DSBs occurring within the same cell. These cells were then left untreated or exposed to ART558, aphidicolin, or both, before being collected for LAM-HTGTS (Fig. 6B). It should be noted that LAM-HTGTS measures the frequency of recoverable (i.e., processed and joinable) ends rather than absolute breakage. As a result, perturbations that alter end processing, for instance, resection/gap-filling and microhomology usage, will change the number of DSBs captured but not necessarily the DNA lesion created.

We found that loss of the polymerase activity of Pol θ reduced the number of captured DSB junctions mapping to RDCs (Fig. 6C, D), while the number of captured DSBs remained higher than with Pol θ-inhibition alone at non-RDC genomic bins (Fig. 6C). We observed that junction losses were associated with a redistribution of LAM-HTGTS junction clusters towards the center of the RDC loci. For example, at the Ctnna2 and Nrg3 loci, the distance between the head-to-head DSB clusters was reduced in aphidicolin-treated cells combined with Pol θ inhibition (Supplementary Fig. 6A). Although previous literature has suggested that the effect of Pol θ would be more pronounced in the G2/M phase, we also observed the reduction in RDC-mapped junction recovery in RDCs within TTR regions (Fig. 6C). At the Large1 locus, the rightward-oriented junction cluster was not recovered (Supplementary Fig. 6A).

We speculated that inhibiting Pol θ polymerase activity would result in reduced micro-homology dependence[34,35] and gap filling[31,36], thereby decreasing DSB end accessibility at RDCs and reducing the number of prey captured by LAM-HTGTS. To test this hypothesis, we analyzed the micro-homology usage at the translocation junction formed between the LAM-HTGTS bait and the endogenous prey DSB ends. We grouped the junctions into two categories: the junction formed between the bait DSB and endogenous DSB lies on a different chromosome (translocations), and a junction formed between the bait and the DSB downstream on the same chromosome (the deletions and inversions) (Fig. 6E and Supplementary Data 3, 5). In Xrcc4/p53-deficient cells, as expected, we observed a preference for two base pair micro-homology usage at the rearrangement junctions (Fig. 6F). In contrast, Pol θ inhibition reverted the micro-homology usage to direct joining, to a level equivalent to that observed in the wild-type cells (Fig. 6F). This is also true for RDC-containing genomic loci (Supplementary Fig. 6C). Hence, the gap-filling defect affects DSB ends genome-wide, including at RDCs.

In addition, while biochemical studies suggested a potential gap-filling activity of Pol θ in vitro[31,32], this has not previously been measured in cycling cells. To examine whether the loss of micro-homology is linked to end processing - including resection and gap filling, we analyzed the total length of the translocated DNA sequence (Fig. 6G, H and Supplementary Fig. 6D and Supplementary Data 4, 5). This analysis was limited to the bait compartment, as the origin of prey prior to end processing was known. For both rearrangements and deletions around the bait sites, we observed a significant and reproducible reduction in the representation of bait length, with an average shortening of approximately 10–15 base pairs. This observation suggested that the polymerase activity of Pol θ either protects the DNA ends from extensive resection, or is required for filling the single-strand DNA gap, thereby creating end-joining-ready DNA breaks.

In summary, in the context of NHEJ-deficient cells, RDC-mediated translocations and recurrent CNVs are regulated by the polymerase activity of Pol θ.

## Pol θ is dispensable for recurrent CNVs in wild-type cells

In order to investigate whether the role of Pol θ in recurrent CNV formation is dependent on NHEJ-deficiency, we performed high-coverage whole genome sequencing experiments for wild-type neural stem and progenitor cells isolated from embryonic mouse brains. We identified four recurrent CNVs (Fig. 7A) in aphidicolin-treated cells,

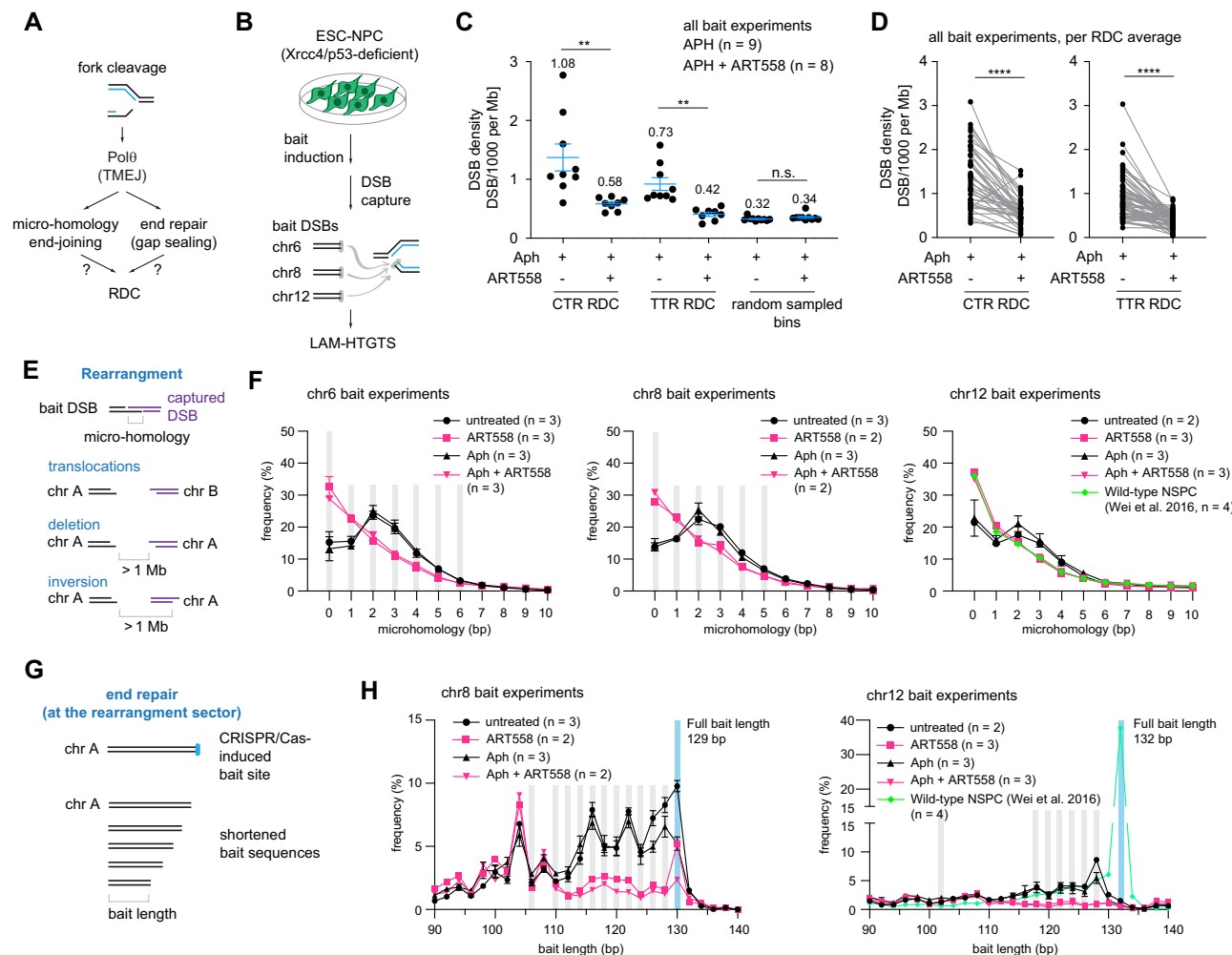

**Fig. 6 | Pol θ activity shapes break formation, capture efficiency and repair pathway choice at RDCs. A** Model for Pol θ function at stalled replication forks. A single-ended break generated at a stalled fork is proposed to reprogram fork structure. Pol θ repairs the broken end and promotes microhomology search, stabilizing RDCs and facilitating theta-mediated end-joining (TMEJ) in Xrcc4/p53-deficient neural progenitor cells. **B** Schematic of the DSB capture assay. CRISPR–Cas9 bait double-strand breaks (DSBs) were introduced on chromosomes 6, 8, and 12 in Xrcc4/p53-deficient neural progenitor cells, followed by treatment with aphidicolin (Aph) and/or the Pol θ inhibitor ART558. Captured DSBs were quantified by LAM-HTGTS. **C** DNA-break density, shown as junctions per thousand per megabase (JPTM), within RDCs located in late constant-timing regions (CTR; 42 RDCs), timing-transition regions (TTR; 65 RDCs), and size-matched randomly shuffled genomic bins (1520 bins). Plots show mean ± SEM from Aph-treated (9 experiments) or Aph + ART558–treated (8 experiments) cells, with mean values annotated. Statistical significance was assessed by two-tailed *t*-test with false discovery rate (FDR) correction (CTR RDC: $P = 5.63 \times 10^{-3}$; TTR RDC: $P = 1.19 \times 10^{-3}$; random sampled bins: $P = 0.191$). **D** Paired before-and-after plots showing changes in DSB density per RDC at CTR (left) or TTR (right). Each dot represents the mean DSB density per RDC from nine (Aph) or eight (Aph + ART558) experiments, with lines connecting the same RDC across conditions. Significance was assessed using a paired two-tailed *t*-test (CTR: $P = 1.0 \times 10^{-10}$; TTR: $P = 1.7 \times 10^{-12}$). **E** Schematic defining rearrangement junctions captured by LAM-HTGTS, including inter-chromosomal translocations and intrachromosomal inversions or deletions within 1 Mb of the bait. **F** Microhomology length distributions at translocation junctions for chromosome 6, 8, and 12 baits, shown as mean ± SEM. Bins with significant differences between Aph and Aph + ART558 conditions (two-tailed *t*-test) are shaded. The number of independent biological replicates for each condition is indicated in the label annotation. **G** Schematic illustrating measurement of bait-end resection at translocation junctions. **H** Distribution of recovered bait-end lengths for chromosome 8 and 12 translocation junctions, shown as mean ± SEM, with significantly different bins shaded as in (**F**). Individual replicate values and junction counts are provided in Supplementary Data 3–5. The number of independent biological replicates for each condition is indicated in the label annotation. Source data are provided as a Source Data file.

three (*Lsamp*, *Lrp1*, and *Csmd1*) of which overlapped with RDCs identified in Xrcc4/p53-deficient neural progenitor cells. We also found that the *Grid2* locus expressed a significant CNV in one of the two repeats. In contrast to Xrcc4/p53-deficient cells, combinatory treatment of aphidicolin and Pol θ inhibition had no effect on recurrent CNV formation (Fig. 7A and Supplementary Data 1), nor did the treatment induce new CNVs. By contrast, Pol θ inhibition alone caused a severe replication-stalling phenotype in which DNA copy-number profiles significantly tracked replication timing (Supplementary Figs. 7A, B), exceeding the magnitude observed in NHEJ-deficient cells. These results imply that the NHEJ pathway inhibits Theta-dependent fork progression.

Since Pol θ inhibition has no effect on recurrent CNV formation, we did not expect changes in DSB density at the RDC loci. Unexpectedly, however, we captured more DNA breaks by using LAM-HTGTS[18] with two chromosomal baits (Fig. 7B, C). We observed a significant increase in RDC-derived junction recovery upon Pol θ inhibition (Fig. 7C) at the *Lsamp* gene locus, the most robust RDC in wild-type neural progenitor cells[18]. We also found gains in the number of detected junctions for the *Grid2* and *Ccser1* loci (Fig. 7C), suggesting a global increase in RDC-mapped junction recovery at RDC hotspots. Indeed, all but two RDCs with multiple recovered junctions exhibited significantly enhanced DSB junction detection upon Pol θ inhibition (Fig. 7D, E), suggesting that

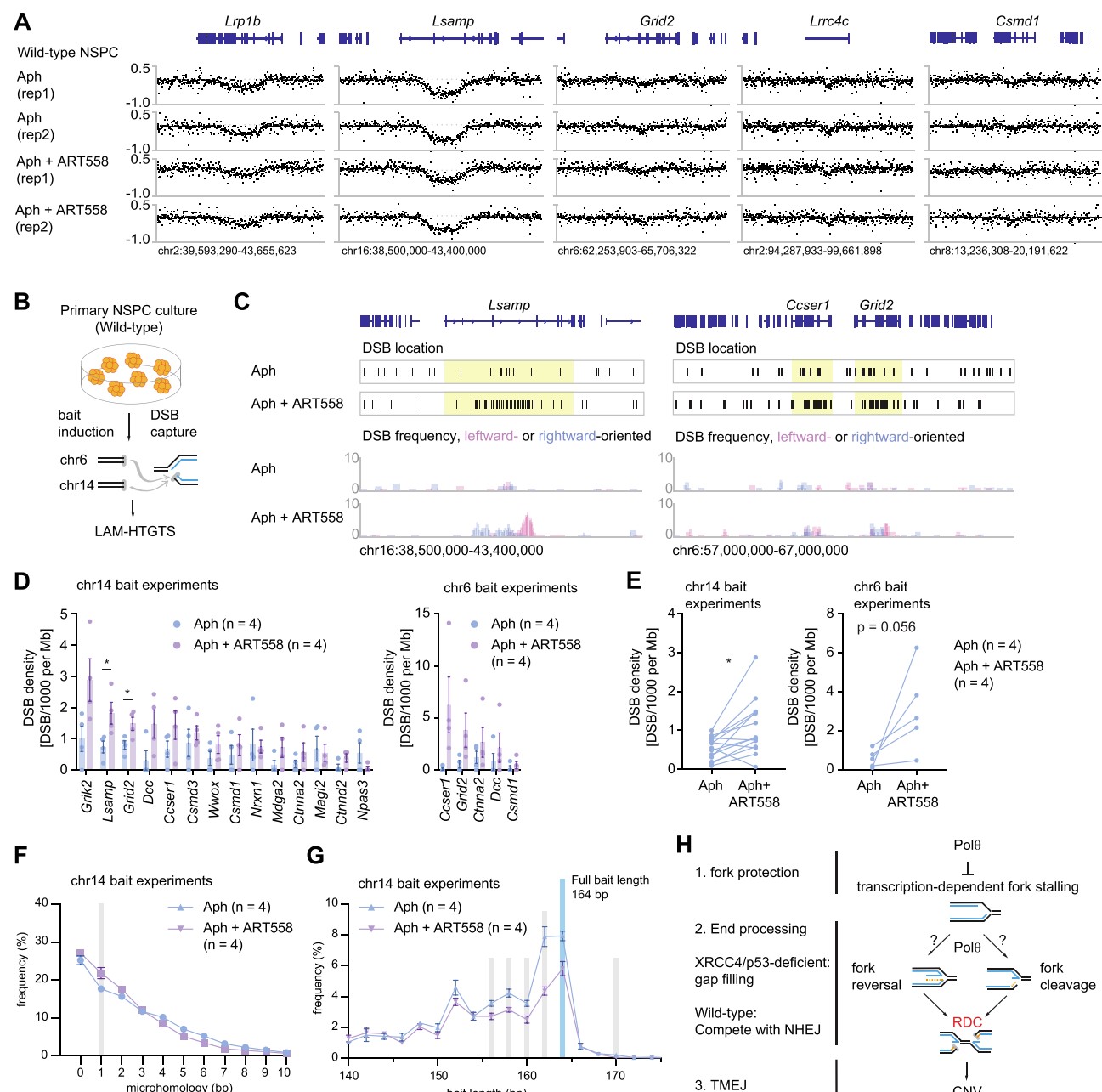

**Fig. 7 | Pol θ inhibition does not alter CNV formation while enhancing DSB density at the RDC loci. A** Whole-genome sequencing coverage ratios at five recurrent CNV loci identified in wild-type neural stem/ progenitor cells (NSPC) following 96 h of the indicated treatments. **B** Experimental diagram. **C** DSB distribution at the *Lsamp* and *Ccser1-Grid2* loci. The exact DSB location and DSB frequencies are shown in separate panels. The corresponding gene loci on the DSB location tracks are highlighted in yellow. 5908 and 6071 total interchromosomal junctions were plotted for Aph and Aph + ART558-treated conditions, respectively. **D** DSB density shown as Junction Per Thousand per Megabase (JPKM) for the RDC hotspots identified in wild-type NSPCs. The figure shows the mean and SEM. Statistical significance was obtained using a two-tailed *t*-test. The figure shows the mean and SEM. P values for *Lsamp* and *Grid2* are 0.035 and 0.035, respectively. Hotspots identified in the chr14 and chr6 experiments respectively are shown here.

**E** Paired DSB density change per RDC in the chr14 and chr6 experiments, respectively. Statistical significance was determined by a two-tailed paired *T*-test. *p = 0.032. **F** Micro-homology frequency at translocation junctions at chromosome 14 baits. The figure shows the mean and SEM. **G** Distribution of recovered bait lengths in deletion-sector reads for chromosome 14. The figure shows the mean (dots) and SEM (error bars). For (**F, G**), values for individual repeat are shown in Supplementary Tables 2–3; junction numbers analysed per experiment, and statistical significance per bin are shown in Supplementary Table 4. **H** A working model for RDC-dependent CNVs. In neural progenitor cells, CNV formation depends on transcription-mediated RDCs. The number of independent biological replicates for each condition shown in Fig. 7D–G is indicated in the label annotation. Source data are provided as a Source Data file.

Pol θ competes with NHEJ to repair DSB ends formed in RDC regions.

We further investigated micro-homology usage and resection status in wild-type neural stem and progenitor cells. In contrast to the Xrcc4/p53-deficient neural progenitor cells, wild-type neural stem and

progenitor cells were less dependent on theta-mediated joining (Fig. 7F and Supplementary Fig. 7C; Supplementary Data 3, 5). Analyzing the total length of LAM-HTGTS bait DNA sequences revealed a slight reduction in the bait length (Fig. 7G and Supplementary Fig. 7D; Supplementary Data 4, 5), less pronounced than that in Xrcc4/p53-

deficient neural progenitor cells, while remaining significant. This indicated that Pol θ−mediated single-strand gap filling appears to occur independently of NHEJ proficiency. In summary, these results suggest that Pol θ activity suppresses the end-repair mechanisms that lead to an extension in DNA break ends. The role of Pol θ in regulating DSB accessibility is context-dependent with respect to RDC formation. In wild-type cells, Pol θ inhibition enhances RDC discovery, potentially by modulating replication timing[37], or through competition with NHEJ. In the absence of non-homologous end joining, neural progenitor cells depend on Pol θ for end processing and the facilitation of microhomology-mediated end joining.

## Discussion

Here, we showed that transcription-dependent RDCs act as the primary initiators of replication stress-mediated CNVs, specifically under aphidicolin treatment. While bulk sequencing identified recurrent focal deletions at late-replicating RDCs, single-cell Strand-seq revealed that RDCs also drive frequent, non-recurrent large-scale losses and complex rearrangements that are invisible to population-based analysis. We propose that the conversion of these breaks into specific structural variants is determined by replication fork orientation−where converging forks favor deletions and unidirectional TTR-associated breaks favor translocations−as well as nuclear architecture and a NHEJ context-dependent reliance on Pol θ.

### RDCs initiate structural variant fixation in post-stress daughter cells

Our findings establish RDCs as major contributors to genome diversification, generating a spectrum of fixed structural variants ranging from focal deletions to large-scale and terminal copy-number losses. Single-cell genome analyses following replication stress demonstrated that recurrent CNVs detected in bulk represent stable structural variants transmitted to daughter cells rather than transient DNA damage. Breakpoints underlying these CNVs were significantly enriched near RDCs, identifying RDCs as primary sources of deletion-initiating DNA breaks. Notably, more than half of aphidicolin-treated single cells exhibited substantial copy-number losses, a feature largely absent in untreated controls, indicating that RDC-associated deletions are frequent rather than rare events. Because large and terminal deletions vary extensively across individual cells, these events are poorly captured by bulk WGS, which relies on recurrence at identical loci. Furthermore, we observed that non-recurrent copy number variations colocalize within RDCs at timing transition regions (TTRs), a substantial number of which involve arm-losses and very large deletions. This may be explained by their genomic positioning outside of nuclear lamina domains, or by their characteristic replication-associated break configuration. Specifically, RDCs in TTRs frequently generate single-ended DSBs with a uniform replication orientation, a structure that does not favor deletions, which require two closely spaced DSBs of opposing polarity. Because such opposing DSB pairs are unlikely to occur at TTR-associated RDCs, CNVs at TTRs are rarely recurrent.

Our findings also imply a direct path from RDC formation to genome evolution. In line with a fitness-based diversification model, terminal chromosomal deletions were frequently observed in aphidicolin-treated cells. Such lesions are well-established triggers of break−fusion−bridge cycles, a potent mechanism for propagating genome instability[38]. We therefore propose that RDC-associated double-strand breaks arising in a prior cell cycle can initiate a replication stress cascade by generating unstable chromosome ends that undergo break−fusion−bridge cycles in subsequent divisions. Supporting this model, we identified a single cell exhibiting hallmark features of a break−fusion−bridge event (Fig. 3F, G), providing direct evidence that RDC-initiated lesions can fuel ongoing structural genome diversification beyond the initial damage event.

It should be noted that while aphidicolin is a well-established reagent for inducing replication stress and has been widely used to map common fragile sites, the inhibition of DNA polymerase alpha may not fully reflect the endogenous forms of replication stress that arise in neural progenitor cells. Distinct replication stressors, such as hydroxyurea, have been shown to produce different CNV patterns[10]. Accordingly, our findings should be interpreted as modeling one specific form of replication stress, rather than the complete spectrum that may occur in vivo.

### Transcription-dependent RDCs are a prerequisite for the formation of CNVs

This study provides experimental evidence that transcription-dependent RDCs induced by replication stress directly generate CNVs. By selectively suppressing transcription at RDC loci, we disrupted RDC formation without altering replication timing, thereby excluding the previously proposed mechanism linking transcription to CNV suppression. Our findings establish a causal relationship, rather than a correlation, demonstrating that a specific subset of CNVs originate from replication stress−mediated DNA breaks triggered by transcription. This work advances current understanding of how genome instability at actively transcribed loci contributes to structural variation. Given that RDCs are DNA replication-dependent, these findings suggest that CNV formation is initiated during S phase. However, due to limitations in our experimental setup, we were unable to directly determine whether CNVs are formed during the S phase or in the G2/M phase, as recently demonstrated[39]. Nonetheless, we cannot exclude the possibility that a subset of S-phase-dependent RDCs persist into the G2/M phase, where they may be subsequently ligated.

It is known that transcription-coupled nucleotide excision repair can leave short ssDNA gaps as repair intermediates[40]. In parallel, topoisomerase activity required for long-gene transcription can introduce ssDNA lesions: topoisomerase I (Top1) processing at embedded ribonucleotides generates a nick and, when trapped as a cleavage complex, Top1cc, a second vulnerable site; when replication forks encounter these gaps, they are converted to single-ended DSBs[41]. The gap-to-break mechanism has primarily been described in early-replicating regions of the genome[42,43]; the transcription- dependent mechanism of RDCs in late-replicating domains warrants further investigation. We speculate that RDC locus accessibility may interfere with the conversion of RDCs to CNVs. Long and late-replicating genes, and heterochromatin, are in spatial proximity with the lamina-associated domains (LADs)[44]. It has been suggested that LADs belong to the B compartment[45], where the close spatial proximity of DNA sequences within a given LAD may enhance the efficiency of end searching required for end joining. In this regard, analyzing RDCs at late CTRs revealed that five out of six genes located within CNVs appeared to associate with LADs (Supplementary Fig. 8). On the other hand, two of the CNV-free, strong RDC-containing loci (*Adgrl3* and *Pcdh9*) are not associated with LADs. In total, for late-replicating RDCs that are not associated with CNVs, only 17 of 57 RDCs are in LADs. In this regard, CNV-containing RDCs were significantly enriched in LADs (chi square test; $p = 0.009$). Yet, the manner in which LADs regulate CNV formation remains to be clarified.

### The polymerase activity of Pol θ promotes RDC end accessibility in the context of Xrcc4/p53-deficiency

In the context of NHEJ-deficiency, we demonstrated that Pol θ promotes aphidicolin-mediated CNV formation at RDC-containing loci. This finding is similar to previous reports in human cell lines[39]. Previous studies exploring the relationship between microhomology-mediated end joining (MMEJ) and NHEJ have provided important insights into the cellular response to DNA double-strand breaks. The prevailing conclusion that Pol θ−mediated repair functions as a backup pathway to NHEJ emerged largely from investigations centered on cell

viability phenotypes and the frequency of selected translocation events[33,46] or copy number variant hotspots at the M phase[39]. In these contexts, MMEJ activity at a genome-wide scale across replication time zones in response to endogenous, cell-intrinsic DSBs was not defined. Our findings expand this framework by demonstrating a broad contribution of Pol θ-mediated repair function in TTR and CTR (Fig. 6C). Following this notion, we demonstrated that the majority of translocations in Xrcc4/p53-deficient cells rely on 2 base-pair microhomology usage (Fig. 6F); inhibiting the polymerase activity of Pol θ diminished the microhomology usage, converting it to the scale of NHEJ. Our results align closely with those of a very recent study focused on an immunoglobulin gene locus in the B-cell genome[35], consolidating that Pol θ contributes to DSB repair across cell cycle phases.

Although biochemical analysis suggested that Pol θ stabilizes the DSB ends[47], this effect has not been measured in cycling cells. We observed a ~ 20-base shortening of bait-derived DNA fragments (Fig. 6H), closely aligning with the biochemical properties characterized in vitro[31,32]. These findings suggest that Pol θ may be required either to convert single-stranded DNA ends into double-strand breaks, or that ART558-mediated inhibition disrupts Pol θ-driven microhomology stabilization. In addition, the end protection ability of Pol θ is independent of NHEJ status, as the same trend was seen in the wild-type neural stem/progenitor cells (Fig. 7G).

### The polymerase activity of Pol θ inhibits RDC end joining in wild-type neural stem and progenitor cells

In contrast to the trend observed in Xrcc4/p53-deficient cells, Pol θ inhibition led to increased RDC DSB end joining in wild-type neural stem/progenitor cells. We observed a marked increase in LAM-HTGTS captured DSBs, such as the Lsamp, Grid2, and Ccser1 gene loci, implying that Pol θ operates to maintain genome stability at these RDC loci (Fig. 6). This finding is consistent with prior knowledge. We speculate that Pol θ protects DNA replication forks, thereby limiting the availability of RDC-derived ends for translocation capture. In this context, Pol θ has been shown to influence replication timing at a subset of genomic loci[37], suggesting it may modulate the replication program in a context-dependent manner. Moreover, Pol θ protects genome stability during S phase and has been implicated in suppressing replication-associated damage outside G2/M phases[48]. In line with this, Pol θ knockout mouse embryonic fibroblasts exhibited nearly double the number of translocation events compared to Ku80/Polq double knockout cells[34]. Taken together, our data suggest that Pol θ constrains RDC-derived junction capture in wild-type neural progenitor cells by safeguarding genome stability during DNA replication.

### NHEJ suppresses Pol θ-dependent DNA replication fork restart

In wild-type neural stem/progenitor cells, Pol θ did not contribute to aphidicolin-induced CNV formation, indicating that intact NHEJ suppresses Pol θ-mediated end joining at stalled replication forks. On the contrary, Pol θ inhibition alone in NHEJ-proficient cells caused more severe DNA content distortions than in NHEJ-deficient cells, suggesting that NHEJ constrains Pol θ-dependent replication progression under replication stress. We propose that sustained Pol θ inhibition activates NHEJ-mediated ligation of DNA breaks, a pathway previously shown to operate at stalled replication forks[49], restoring DNA continuity but leaving single-stranded nascent gaps unresolved, thereby preventing replication restart and effectively freezing replication intermediates. In contrast, in the absence of NHEJ, unrepaired breaks persist at stalled forks, allowing exposed single-stranded DNA, particularly on the leading strand, to engage recombination-based restart pathways, including single-strand annealing between sister chromatids. Consistent with this model, combined loss of Lig4 and Pol θ has been shown to enhance single-strand annealing–mediated repair events[50], supporting a pathway switch from end joining to homologous

mechanisms when both Pol θ and NHEJ are compromised. In addition, we speculated that persistent aphidicolin exposure generates extensive nascent single-stranded DNA gaps[51], which exceed the repair capacity of Pol θ, thereby forcing a switch to Pol θ–independent fork restart or remodeling pathways.

### Implications in neuronal disorders and cancer

Somatic CNVs in the brain have been widely linked to both neuropsychiatric disorders and cancer[5,52], with replication stress acting as a major driver of CNV formation. During prenatal and early postnatal development, the brain undergoes a period of pronounced cellular proliferation to generate the vast neuronal architecture required for mature function. This intense replicative demand renders neural progenitor cells particularly susceptible to replication stress, creating a window of vulnerability for genomic instability[18]. It has been hypothesised that CNVs may disrupt synapse formation by generating novel protein isoforms or deleting critical gene segments[18]. Given that many RDC-containing genes encode proteins involved in synaptic function, the resulting genetic heterogeneity, particularly in cell adhesion molecules, could impair the establishment of stable synaptic interactions[53]. In addition, replication stress-induced CNVs have been previously proposed as a key driver of structural chromosomal instability in tumors[54]. For instance, the RDC-containing genes: LRP1B, NPAS3, LSAMP, and SMYD3 have all been found to be frequently deleted in glioblastoma[55,56]. Furthermore, large-scale CNV signature analyses across human cancers identified specific CNV patterns related to replication stress[5]. Notably, signature CN9, characterized by loss of heterozygosity and linked to TP53 mutations in glioblastoma, may arise through a similar replication stress-dependent mechanism reported by us. Our study provides a mechanistic basis that could help explain how such CNV signatures emerge under conditions of replication stress, with broader implications for tumor evolution.

## Methods

### Mouse embryonic stem (ES) cell-derived neural progenitor cells (NPC)

Mouse embryonic stem (ES) cell-derived neural progenitor cells (NPCs) deficient in both Xrcc4 and p53 (Xrcc4−/−p53−/−) were used in this study, as previously described[20]. The Xrcc4−/−p53−/− ES cells were cultured in DMEM supplemented with 15% ES cell-qualified fetal bovine serum, 20 mM HEPES, non-essential amino acids, 100 U/ml penicillin-streptomycin, 2 mM glutamine, 0.1 mM β-mercaptoethanol, and 1000 U/ml ESGRO recombinant mouse leukemia inhibitory factor (LIF, Millipore ESG1107). The cultures were maintained on a monolayer of irradiated mouse embryonic fibroblasts.

Differentiation of the Xrcc4−/−p53−/− ES cells into NPCs was performed following previously established protocols[20,57]. Briefly, the ES cells were seeded onto plates coated with laminin and poly-L-ornithine (Sigma Aldrich, P4967) and cultured in N2B27 medium, consisting of 50% DMEM/F12 (ThermoFisher Scientific, 11330057), 50% Neurobasal (ThermoFisher Scientific, 21103049), 1% modified N2 supplement (Gibco, 17502048), 2% B27 supplement without retinoic acid (ThermoFisher Scientific, 12587-010), and 1× Glutamax (ThermoFisher Scientific, 35050061) for seven days. Subsequently, the cells were passed onto laminin-coated plates and maintained for an additional 5–6 days in NBBG medium. This medium consisted of Neurobasal-A (ThermoFisher Scientific, 10888-022) supplemented with 2% B27 without retinoic acid, 0.5 mM Glutamax, 10 ng/ml human epidermal growth factor (EGF, ThermoFisher Scientific, PHG0314), and 10 ng/ml mouse basic fibroblast growth factor (FGFb, ThermoFisher Scientific, PMG0034).

### Wild-type, primary neural stem and progenitor cell culture

The organ and embryo collection were covered under an institutional internal animal license (DKFZ381). All mice were bred in-house at the DKFZ Center for Preclinical Research and maintained under specific

pathogen and opportunistic-free conditions on a 12-h light-dark cycle in a temperature-controlled environment with *ad libitum* access to food and water. Pregnant C57BL/6 J female mice were sacrificed after 17 days of a positive vaginal plug. Embryonic day (E) 17.5 embryos were retrieved, decapitated, and the brains were isolated and dissociated into a single-cell suspension. In brief, brains were chopped into 2 mm pieces, followed by papain digestion. Single cells were passed through a Percoll gradient, washed with DPBS and NBBG, and were plated at a density of 1 million per mL in ultra-low-attachment six-well plates (Merck, CLS3471). Fresh growth factors (EGF, FGFb, as described above, and PDGF, ThermoFisher Scientific, PMG0044) were added to the culture medium every second day. Five days after seeding, the neural spheroids were dissociated with TrypLE select (ThermoFisher Scientific, 12563029), washed in DPBS and NBBG, and seeded to fresh ultra-low-attachment six-well plates at the same cell density. Five days after seeding, we dissociated the cells and performed nucleofection as described in the LAM-HTGTS section below.

## Whole genome sequencing

**Library preparation.** Approximately $5 \times 10^6$ *Xrcc4−/−p53−/−* ES cell-derived NPCs were seeded onto freshly prepared poly-L-ornithine and laminin-coated plates and subsequently treated with either DMSO or 0.5 μM APH for a total of 96 h. The treatment medium was refreshed after the initial 48 h. Following treatment, cells were harvested, and genomic DNA (gDNA) was extracted with a standard phenol/chloroform/isopropyl protocol. Libraries were prepared following a TruSeq Nano DNA protocol by the NGS core facility staff at the DKFZ. Libraries were sequenced using an Illumina NovaSeq 6000 platform with 150 bp paired-end reads, achieving an average sequencing depth ranging between 160× and 200× per sample.

## Alignment, QC, and CNV calling

Post-sequencing, the raw FASTQ reads were subjected to the automated AlignmentAndQC workflow (https://github.com/DKFZ-ODCF/AlignmentAndQCWorkflows/) with alignment to the mm10 genome build. $\log_2$(APH/DMSO) values were calculated using deepTools' *bamCompare* function. To account for differences in sequencing depth between Aph- and DMSO-treated samples, we applied the built-in scale factor option. The accuracy of this normalization was verified by independently calculating the scale factor as the ratio of the average sequencing depth of APH to that of DMSO.

To quantify copy number variants across RDC-containing genomic regions, the $\log_2$ copy number ratio data $\log_2$(Aph/DMSO) was analyzed using a custom R script. The script imported $\log_2$ ratio data in bedGraph format and genomic regions of interest (in this case RDCs) in BED format. Non-standard chromosomes were filtered out, and the data were converted into GRanges objects. Overlapping bins between the $\log_2$ ratio dataset and RDC regions were identified using the findOverlaps() function from the *GenomicRanges* package. For each RDC region, the average $\log_2$ ratio was calculated across all overlapping bins. Results were exported as a tab-delimited text file.

To assess statistical significance, 10 sets of shuffled RDC regions were generated. These shuffled regions maintained the original lengths of the RDCs but were randomly repositioned across the mouse genome (mm10), ensuring chromosome sizes were respected. Each shuffled set was saved as a separate BED file for downstream analysis.

We also applied Delly2 for CNV calling. Four customized scripts were used for this purpose. Briefly, CNVs were identified from whole-genome sequencing data of Aph-treated and DMSO-treated samples. Read-depth profiles were generated for both Aph and matched DMSO BAM files, followed by SCNA calling using the mm10 reference genome and a predefined mappability blacklist. Somatic copy number alteration (SCNA) calls were merged and filtered for APH-specific events based on sample classification (Aph vs. DMSO), using thresholds including segment size ≥1000 bp and ploidy level ≥2. Segmentation

files were extracted and visualized. SVs were jointly called from Aph and DMSO BAMs, applying an exclusion list to mask problematic regions, and filtered for Aph-specific variants with a minimum mapping quality of 50. To improve somatic specificity, pre-identified variant sites were genotyped across a panel of unrelated DMSO controls, and Aph-specific SVs were retained based on consistent classification using updated sample metadata. The final filtered variants were annotated against the mouse mm10 gene model using a GTF annotation file.

## Strand-seq sample and library preparation

Single-cell Strand-seq libraries were prepared according to the OP Strand-seq protocol[58] with the following modifications: A total of $3 \times 10^6$ ES cell-derived neural progenitor cells (NPCs) were cultured for 96 h in the presence or absence of aphidicolin (Aph). For Aph-treated conditions, cells were exposed to 0.5 μM Aph for 96 h. Aph was then washed out, and cells were cultured for a further 96 h, followed by 40 μM BrdU + 1 μM 5-FdU (Sigma, B5002, Sigma343333) for 18 h in control cells and 21 h in Aph-treated cells. Viable cells were counted, pelleted, and resuspended at $1 \times 10^7$ cells per mL in ice-cold lysis buffer (LB; 15 mM Tris-HCl (pH 7.5), 10 mM NaCl, 80 mM KCl, 2 mM EDTA, 0.5 mM EGTA, 0.1% (v/v) 2-mercaptoethanol, 0.1% Triton X-100, 0.2 mM spermine, and 0.5 mM spermidine.). After 15 min on ice, nuclei were collected ($250 \times g$, 4 min, 4 °C), washed three times in ice-cold H4-10 buffer (4 mM HEPES pH 7.5, 10 mM CaCl$_2$, 0.05% Tween-20), and treated with RNase A (0.1 mg/mL) at 37 °C for 15 min. Aliquots of $1 \times 10^6$ nuclei in 100 μL H4-10 were digested with MNase (NEB M0247) at 37 °C for 15 min, then fixed in 3% formaldehyde for 15 min at room temperature, mixing every 5 min to maintain nuclei in suspension. Fixed nuclei were pelleted ($250 \times g$, 4 min, 4 °C) and resuspended in ice-cold LB to derive $10^7$ nuclei per mL[24,25,59]. Single nuclei were sorted into 3 μl of release buffer[58], spun down and frozen at −80 °C until Strand-seq library production. On the day of Strand-seq library production, the plate was placed on ice to allow slow thawing. All further processing was done on an i7 Biomek Beckman liquid handling station. Decrosslinking, protease digestion, polishing and adapter ligation were all done exactly as described in the original OP Strand-Seq protocol with only the necessary adjustments to volumes based on the starting volume difference. Our protocol also included a bead-based cleanup after adapter ligation performed at 0.8x bead ratio to get rid of most of the adapter dimers prior to PCR amplification. Following this cleanup, DNA was exposed to Hoechst and UV as per OP Strand-seq protocol and finally libraries were amplified for 15x PCR cycles using iTru dual indices. Raw reads were aligned to mm10 genome build. Structural variant detection was performed using MosaiCatcher v2[60] and Strand-Tools[27].

## Strand-seq Analysis

**CNV analysis in single cells.** StrandTools[27] was applied to the experimental data to generate strand-specific genomic profiles for all cells, with the genome partitioned into 200 kb bins. Automated quality control was performed at the level of individual genomic segments, and only segments passing QC, first through ASHLEYS[61] followed by manual validation, were retained for downstream analyses. Strand inheritance was then assigned for each segment as Watson-only, Crick-only, or mixed Watson/Crick. Only mixed-orientation segments were retained for copy-number variant calling, as this configuration allows strand-specific coverage to be used as an internal reference, enabling discrimination between true copy-number loss and strand-specific coverage dropout.

CNV calling was performed by aggregating all 200 kb bins overlapping each gene. Genomic regions exhibiting signal on only one strand (Watson or Crick) were assigned a copy number of one, whereas regions with significant signal on both strands were assigned a copy number of two. Regions lacking signal on both strands were excluded,

as complete copy-number loss cannot be distinguished from global coverage depletion (Supplementary Fig. 3A). One cell, which did not contain any bins in the Watson and Crick orientation in the genes of interest was excluded from the analysis. For the CNV analysis carried out across the whole genome, the mean CNV was taken across all Watson and Crick oriented genomic segments.

### Breakpoint analysis

Breakpoint Identification and Resolution: breakpoints were identified from single-cell Strand-seq data. Due to the inherent resolution of Strand-seq, which is constrained by the density of template-strand transitions, each breakpoint coordinate was expanded symmetrically into a 2 Mb window. This windowing approach accounts for the technical uncertainty in single-cell mapping and ensures that the biological lesion is captured within the local genomic context.

Genomic Intersection and RDC Harmonization: recurrent DNA-break clusters (RDCs) were defined as GRanges objects using the mm10 reference genome. To avoid statistical inflation, we utilized a binary overlap logic: a breakpoint window was recorded as a single hit if it intersected an RDC.

Clustered Permutation Testing: to assess the significance of the association between breakpoints and RDCs, we employed a hierarchical permutation strategy using the RegioneR framework. This clustered approach was used to account for inter-cell variance.

- Observed Statistic: The total number of unique breakpoint-window hits was summed across the entire population of analyzed cells.
- Null Model: To construct a biologically realistic random expectation, 10,000 permutations were performed where breakpoint windows were shuffled exclusively within their original chromosomes, and within their specific cell of origin. By maintaining this per-cell and per-chromosome structure, the null model accounts for the non-uniform distribution of structural variants across the genome.
- Significance Assessment: A one-sided p-value was calculated by comparing the observed total hits against the permuted distribution. Standardized Z-scores were used to measure the magnitude of enrichment relative to the random null expectation.

### Generating promoter-enhancer deleted cell lines for the *Ctnna2* and *Nrg3* loci

To generate *Ctnna2* and *Nrg3* promoter-enhancer-deleted (pe-del) ES cell clones, CRISPR/Cas9-mediated genome editing was performed using pairs of single guides RNAs (sgRNAs) designed to flank target regions. For *Ctnna2*, sgRNAs were positioned upstream of the cis-regulatory element and downstream of the final exon to generate *Ctnna2* deletion clones, and flanking the entire *cis*-regulatory region to produce pe-del clones. For *Nrg3*, two sgRNAs were selected—one upstream and one downstream of the cis-regulatory elements—based on design and efficiency evaluation using ChopChop[62]. In both cases, cis-regulatory elements were defined based on histone modification marks (H3K4me, H3K9ac, H3K27ac) using ENCODE 3 ChIP-seq data from the UCSD/Ren lab. Individual ES cell clones were isolated, expanded, and screened by genotyping PCR to identify those carrying the intended deletions. Positive clones were further validated by Sanger sequencing to confirm precise deletion events.

### Two-fraction replication sequencing (Repli-Seq)

Two fractions of Repli-seq were generated and analyzed as described[63]. In brief, ES cell-derived neural progenitor cells were plated at 30% confluence three days before BrdU incorporation. Cells were treated with 100 μM BrdU (Sigma, B5002) for 2 h, and cells were washed with cold DPBS ten times to remove loosely attached and dead cells. Cells were detached with Accutase, spun down, and resuspended in 2.5 ml

ice-cold PBS plus 1% (vol/vol) FBS. Cells were fixed by adding 7.5 ml ice-cold 100% ethanol dropwise. The fixed cells were either stored at −20 °C until sorting or directly stained with propidium iodide. Cell pellets were washed with ice-cold 1% (vol/vol) FBS in PBS once and resuspended in 0.5 ml of PBS/1% (vol/vol) FBS/propidium iodide (Sigma-Aldrich, P4170)/RNase A (Thermo Scientific, EN0531) solution as described. The final cell suspension concentration should be $4 \times 10^6$ cells/ml. Cells were filtered through a 35 μm nylon mesh and sorted using FACS. During the sorting, early or late replicating ES cell-derived NPCs were separated into two fractions; for each early and late fraction, 120,000 cells were sorted. The cells were pelleted, and genomic DNA was extracted as described. DNA was fragmented with a Covaris S220 Focused-ultrasonicator to 200-bp average fragment size, and BrdU-positive DNA was enriched using anti-BrdU antibody. The antibody was purchased from Santa Cruz biotech, cat. no. sc-32323-ac. Libraries were constructed using the NEBNext Ultra DNA Library Prep Kit (New England BioLabs, E7370L) before BrdU pulldown, indexing, and amplification. Before submission, the libraries were further purified with AMPure XP beads (Beckman Coulter, A63881) and underwent quality control. After pooling, libraries were sequenced on Illumina NextSeq (75 bp single-read) or NovaSeq X-plus (100 bp paired-read). After sequencing, the adaptors were trimmed from the raw FASTQ reads, and the reads were aligned to mm10/GRCm38 through Bowtie2 using a singularity container (https://github.com/brainbreaks/HighRes_RepliSeq). The under replication score was determined by the Repli-Seq R package (https://github.com/CL-CHEN-Lab/RepliSeq) using the calculateURI function[14], with the two-fraction Repli-Seq datasets.

### Linear amplification-mediated, High-throughput, genome-wide translocation sequencing (LAM-HTGTS)

**Library used in this article.** For comparing RDC and CNV positions (Fig. 2), we extracted the published data GSE233842 generated from the *Xrcc4−/− p53−/−* ES cell-derived NPCs. We produced 89 primary LAM-HTGTS libraries for experiments shown for Figs. 6 and 7, which were deposited into NCBI's GEO repository with the accession number GSE305347 [https://www.ncbi.nlm.nih.gov/geo/query/acc.cgi?acc= GSE305347].

**Library preparation.** To induce targeted bait sites on chromosomes 6, 8, 12, or 14 in *Xrcc4−/− p53−/−* ES cell-derived neural progenitor cells or wild-type NSPCs, $5 \times 10^6$ cells were nucleofected with 5 μg of SpCas9/sgRNA expression plasmids (pX330-U6-Chimeric_BB-CBh-hSpCas9; Addgene plasmid #42230). SgRNA sequences specific to each bait site were individually cloned into pX330 vectors following the protocol described by the Zhang lab (https://www.addgene.org/crispr/zhang/). Nucleofection was performed using the Nucleofector 2b device (Lonza) with the Mouse NSC Nucleofector Kit (VPG-1004, Lonza) according to the manufacturer's instructions.

Following nucleofection, cells were cultured for 96 h under treatments. For aphidicolin-treated conditions, cells were treated with 0.5 μM aphidicolin for 72 h, followed by an additional 24 h at 0.25 μM. For polymerase theta inhibition, cells were treated with 5 μM inhibitor for 72 h, followed by an additional 24 h at 2.5 μM. Genomic DNA was then extracted, and sequencing libraries were generated using established protocols. Libraries were sequenced on an Illumina NextSeq platform with 150bp paired-end chemistry.

### Demultiplexing and alignment

Sequencing reads from demultiplexed FASTQ files were processed using the HTGTS Docker container (https://github.com/brainbreaks/HTGTS). The *TranslocProcess* module was used to trim adapters and perform library-specific demultiplexing. Trimmed reads were then aligned to the mm10/GRCm38 mouse reference genome using

Bowtie2 and further processed using the *TranslocWrapper* function integrated into the pipeline. The file *result.tlx containing the bait and the prey information for each library was used for downstream analysis.

### DSB density, micro-homology usage, and bait-length analyses

To calculate the number of DSB to bait junctions detected in pre-specified regions-of-interest, we extracted the DSB coordinates from the *result.tlx files. All junctions were used to plot the junction density in Fig. 2. For calculating the DSB density in the following figures, only DSBs identified at the non-bait chromosomes were considered. We used bedtools to count the number of recovered prey junctions per RDC in each LAM-HTGTS library. The number of recovered junctions per RDC was normalized using the total non-bait chromosomal DSBs, multiplied by a thousand, and divided by the length of RDC, in Megabase. This calculation derives DSB density as Junction per Thousand per Megabase, which is shown for Figs. 6 and 7.

To calculate microhomology (MH) usage, we extracted information from the LAM-HTGTS output files. We extracted $B\_Qend$, which represents the length of the bait which translocates to an endogenous DSB, and $Qstart$, which represents the position where the endogenous DSB starts on the same read. Translocation junctions containing microhomology between 0–10 bp were kept for the calculation. MH was shown as frequency for Figs. 6 and 7. The length of MH was deduced by

$$MH = -(Qstart - B\_Qend - 1) \qquad (1)$$

To calculate bait length (Blen), we calculated the distance between the beginning of the nested LAM-PCR primer (PrimStartCo) and the observed bait end or start ($B\_Rend$ or $B\_Rstart$ respectively) per read. All reads containing a junction were counted, and the frequencies of bait lengths are shown for Figs. 6 and 7.

In case of a bait DSB pointing at a *plus* orientation,

$$Blen = (B\_Rend - PrimStartCo + 1) \qquad (2)$$

In case of a bait DSB pointing at a *minus* orientation,

$$Blen = (PrimStartCo - B\_Rstart + 1) \qquad (3)$$

### Global run-on sequencing (GRO-seq)

GRO-seq libraries were prepared as described[16,28]. For each GRO-seq experiment, 5 – 10 million ES cell-derived neural progenitor cells' nuclei were isolated for global run-on analyses. Two technical replicates per experiment condition were performed. GRO-seq FASTQ files were aligned to the genome build mm10/GRCm38 through Bowtie2 and processed using the GRO-seq pipeline (https://github.com/brainbreaks/GROseq).

### Reagents and laboratory equipment

Chemicals, cell culture reagents, solvents, primers and oligo sequences, antibodies, as well as specific laboratory equipment used for this article are presented in Supplementary Data 6.

### Reporting summary

Further information on research design is available in the Nature Portfolio Reporting Summary linked to this article.

## Data availability

The whole genome sequencing data generated in this study have been deposited in the European Genome Archive database under accession code PRJEB95922. The two-fraction Repli-Seq and LAM-HTGTS generated in this study have been deposited in NCBI's Gene Expression Omnibus and are accessible through GEO Series accession number GSE305347. The GRO-seq data generated in this study have been deposited in NCBI's GEO database under accession code GSE305346. Processed data were aligned to mm10 genome build. The Strand-seq data have been deposited in the European Genome Archive database under accession code PRJEB105360 [https://www.ebi.ac.uk/ena/browser/view/ PRJEB105360]. We retrieved high-resolution Repli-Seq data generated in this study from GSE257765 (related to Fig. 2), LAM-HTGTS data from GSE233842 [https://www.ncbi.nlm.nih.gov/geo/query/acc.cgi?acc= GSE233842] (related to Fig. 2) and GSE74356 (wild-type Chr12 data, Fig. 6; re-aligned to mm10) for replotting. Source data are provided with this paper.

## Code availability

In this manuscript, three singularity containers were applied for HTGTS (https://doi.org/10.5281/zenodo.10843397), GRO-seq (https://doi.org/10.5281/zenodo.10838367), and Repli-seq (https://doi.org/10.5281/zenodo.10838365) analyses. In addition, an automated AlignmentAndQC workflow (https://github.com/DKFZ-ODCF/AlignmentAndQCWorkflows/) was applied to WGS analysis. Bowtie 2 2.5.0 used for LAM-HTGTS, Repli-seq, and GRO-seq. Samtools 1.7 was used for LAM-HTGTS, and 1.8 and above was used for Repli-seq and GRO-seq. These pipelines and software dependencies were containerized using Docker, ensuring reproducible and consistent version control across analyses. Delly 1.2.6 is a publicly available package (https://github.com/dellytools/delly). MosaicCatcher V2 (https://github.com/friendsofstrandseq/mosaiccatcher-pipeline) and Strandtools (https://git.embl.org/cosenza/strandtools) are publicly available.

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

## Acknowledgements

This work is supported by the Helmholtz Young Investigator grant, a Life and Health Alliance seed grant program, and a European Research Council (ERC) starting grant (BrainBreaks, grant no. 949990) to P.-C.W. Additional funding for this work came from the ERC Advanced grant (SEE-MAGIC, grant no. 101098056) to J.O.K. We thank Marco Giaisi for managing the Wei lab. We also thank the FACS core facility, the NGS core facility, the ODCF data management core facility, and the staff at the preclinical center at the DKFZ. We also thank the Korbel lab for fruitful discussions and suggestions. We extend our acknowledgements to Pavel Janscak and Jan Korbel for their valuable feedback. We also thank the team members at the Wei lab for intellectual discussions and technical support.

## Author contributions

L.C. and P.-C. W. conceptualized and designed the project. L.C., V.I., E.B., P.H., A.M., N.T., G.D.M., B.D., J.B., N.C. and M.G. performed the experiments: WGS (L.C. and N.C.), ES cell pe-del clones (V.I., A.M., and N.T.), Repli-Seq (L.C. and B.D.), GRO-seq (L.C. and V.I.), Strand-seq (L.C., E.B., and P.H.), and LAM-HTGTS (L.C., G.D.M., B.D., and M.G.). A.I., M.R.C., T.W., S.B., L.C., and P.-C.W. analyzed the data. L.C., A. I., V. I., M.R.C., J.O.K., A. M., and P.-C.W. interpreted the results. A.I., L.C., and P.-C.W. created figures and wrote the manuscript. T. H. supervised team member (S.B.). P.-C.W. secured research budget, supervised team members, and oversaw the research.

## Funding

## Competing interests

J.O.K. has previously disclosed a patent application (no. EP19169090) that is relevant to the use of Strand-seq for somatic structural variation analysis. The remaining authors declare no competing interests.
