## [Transparent Peer Review file · Nature Communications]

Recurrent DNA Break Clusters Drive Replication-Stress-Induced Copy Number Variants and Genome Diversification

Corresponding Author: Dr Pei-Chi Wei

Version 0:

Reviewer comments:

Reviewer #1

(Remarks to the Author)

In this manuscript, Corazzi et al. identify an association between somatic CNVs and recurrent DNA break clusters in aphidicolin-treated murine neural progenitor cells. They then go on to show the involvement of DNA polymerase theta in the formation of recurrent DNA break clusters. This was an interesting and well-presented manuscript that was enjoyable to read. My comments for the authors are below.

Points that I request the authors address and respond to:

- I understand the authors' choice of aphidicolin to induce DNA replication stress, as there is very good historical precedent for doing so (in, for example, the detection of common fragile sites). However, it strikes me that the inhibition of DNA polymerase alpha induces a type of replication stress that may not mirror the type of replication stress that could result in CNVs in neural progenitor cells in the absence of any agent. This is particularly pertinent given that replication stress induced by hydroxyurea and aphidicolin generate distinct patterns of somatic CNVs (<https://doi.org/10.1186/s13059-022-02781-0>). I am certainly not suggesting that the authors repeat the analysis with different agents that induce replication stress; that would be quite unreasonable. However, this would be an interesting point to discuss, and it would be helpful to qualify the conclusions appropriately.
- Figure 1B shows the rank-ordered $\log_2(\text{Aph}/\text{DMSO})$ coverage for every RDC for each of two replicates. While the shape of the curve shows good agreement between replicates, it is not clear to me from this plot whether the rank order itself is well-preserved across replicates. Figure 1C suggests the agreement is generally good, but this would still be nice to show.
- At the opening of the discussion, the authors state, "In this study, we establish that transcription-dependent RDCs generated under replication stress directly cause somatic CNVs at ultra-long, late-replicating neuronal genes." This feels a bit strong given the numbers involved: Of the nine CNVs called by Delly2, six of them mapped to an RDC with two not being reproducible across replicates. However, I think there are some fairly straightforward analyses that would help the authors strengthen their conclusions. For instance, what is the probability that the size of CNVs observed overlaps an RDC by random chance? Likewise, it would help if the authors could state how many genes they consider "ultra-long, late-replicating neuronal genes" so that we can get a feel for how many of them show this concordance between CNVs and RDCs.
- In Figure 5C-D, it is important to see exact p-values to evaluate the results appropriately. If I understand the figure legend correctly, those genes that are significant are hotspots, but based on the other genes and the shape of the distributions, it seems as though these p-values might be quite close to 0.05.

Minor points that should be straightforward to address:

- Line 90-91: A fraction of RDC-containing genes *are* copy number variation hotspots in neural progenitor cells
- Line 652: false *discovery* rate (FDR) correction
- Some readers may find the use of "constant-timing region" (CTR) confusing. First, clearly defining what a CTR is in the main text would make it more accessible to nonspecialists. Second, there is a risk of confusion in the legend of Figure 2 (Lines 615-616) where the grammar suggests that a CTR is a region whose timing is unchanged by aphidicolin treatment, but this is not actually what a CTR is nor is it what the authors mean. #
- For me, I found the light and dark blue a bit difficult to distinguish between in Figure 1D.
- For reproducibility, it is useful to specify the exact versions of any software used wherever possible. E.g., Bowtie2, Delly2, etc.
- Having looked through <https://github.com/brainbreaks/>, it's not completely obvious to me what was and was not used for this manuscript. It looks like there's more here than was used, and it would help to make the Code Availability statement

more specific. That said, the repositories look very nicely documented which is highly commendable. All of the pipelines here look very readable and usable.

Reviewer #2

(Remarks to the Author)

In this manuscript Corazzi et al. focus on the role of POL θ in the replication-stress-mediated generation of copy-number variations at recurrent DNA break clusters. The origin of these events and their implications for neurological disease and cancer is an intriguing question in the genome instability field. I have, however, reserves regarding the conceptual contribution of the current manuscript. The manuscript shows:

- An association between transcription-dependent RDCs induced by replication stress and CNVs. This is already established by several publications, as cited in the manuscript.
- The involvement of POL θ in generating RDC-associated translocations in NHEJ-deficient background. It is well-known that POL θ -mediated repair (by MMEJ) acts as a backup for NHEJ. The contribution of showing that this also occurs in this particular context is minor.
- The involvement of POL θ in preventing RDC-associated translocations in wild-type background. This is the most interesting observation, which the authors explain by a potential role of POL θ in preventing the emergence of replication-associated DSBs. Although this is not completely novel, as the authors indicate, it would be worth exploring further.

Therefore, in my view, the manuscript is confirmatory and preliminary, having made an interesting observation that would need to be studied in more detail to provide novel insights that can constitute a true advancement to our mechanistic knowledge on the sources of RCD-associated CNV.

Reviewer #3

(Remarks to the Author)

To investigate the relationship between replication stress-induced DSBs and CNVs, Corazzi et al. employed ultra-deep whole-genome sequencing to identify CNV hotspots upon replication inhibitor treatment in neural stem and progenitor cells. Building on their previously defined recurrent DNA break clusters (RDCs), they observed that a subset overlapped with Aph-induced CNV hotspots and that these RDC/CNV hotspots were transcription-dependent. They further examined the contribution of Pol θ , an MMEJ factor, to CNV generation in both NHEJ-deficient and WT contexts. Collectively, they conclude that RDCs may represent a potential source of CNVs.

My main concern, however, is the overall prevalence of these RDC-induced CNVs. Are they relatively common or rather rare events? Specifically, the manuscript does not provide the total number of genome-wide CNV hotspots; instead, the analysis is restricted to RDC-related CNVs. It would be important to report the proportion of RDC-associated CNVs relative to all CNVs detected. Moreover, in their previous study (Nat. Commun. 2024), the authors identified 152 RDCs, most located at TTRs, CTRs, and IZs, which generally do not overlap with CNV hotspots. Only 5 out of 152 RDC loci were reported to exhibit CNV formation. This raises the question of whether the contribution of RDCs to CNV generation is quantitatively minor, and further clarification and contextualization would strengthen the manuscript.

Other comments:

1. The WGS experiments were performed in cells treated with 0.5 μ M aphidicolin (Aph) for 72 h followed by 0.25 μ M Aph for 24 h. The rationale for changing the dose after 72 h is unclear. Could this treatment regimen primarily generate under-replicated regions rather than bona fide CNVs? Performing WGS in G1-arrested cells after Aph treatment would help address this concern.
2. Some of the interpretations appear overstated. For example, the conclusion that Pol θ suppresses RDC-prone hotspots is not supported by direct evidence. Similarly, they showed the CNVs/RDC are transcription dependent at two sites, then conclude "DNA sequence loss in CNV hotspots is RDC-dependent", which could be correlative than causal.
3. No data directly demonstrate the effect of Pol θ inhibition on CNV formation. The influence of Pol θ at RDCs is described as general, without distinguishing features specific to CNV-associated RDCs.
4. The manuscript does not address why only a subset of RDCs give rise to CNVs while the majority do not. Clarification or speculation on this discrepancy would be valuable.
5. Certain statements are confusing or imprecise. For instance, the subtitle "RDC accessibility" is unclear, and the phrasing "Six of these deletions mapped directly within an RDC showing significant DNA sequence losses" is ambiguous, does this refer to six distinct RDCs or six deletions within a single RDC?

Version 1:

Reviewer comments:

Reviewer #1

(Remarks to the Author)

I feel the authors have addressed all of my comments, and I thank them for making the changes very easy for me to see and review in the main text. Supporting the manuscript with singularity images of the pipelines is particularly commendable, and I appreciate the effort put into making these accessible.

Reviewer #2

(Remarks to the Author)

The authors have undertaken substantial experimental work and text clarification to address the major concerns raised during the initial review, including new single-cell analyses and additional genomic experiments that significantly strengthen the causal link between RDCs and structural variation. While some mechanistic questions remain open for future work, they do not detract from the central conclusions, which are now well supported by the data. I therefore support publication.

Reviewer #3

(Remarks to the Author)

The authors have addressed my questions.

Response to Reviewers (NCOMMS-25-64474A)

Corazzi L. et al., Recurrent DNA Break Clusters Drive Replication-Stress-Induced Copy Number Variants and Genome Diversification.

Remark for all reviewers:

We thank the reviewers for their encouraging comments and constructive suggestions, which have led to significant changes and clarifications that have considerably improved the manuscript. We have now performed two additional sets of experiments to: (1) Examine the contribution of recurrent DNA break clusters (RDCs) to non-recurrent copy number variant (CNV) frequency (2) Determine the role of DNA polymerase theta (Pol θ) in CNV formation. We have also carried out a range of additional statistical analyses and textual clarifications to bolster the claims made in the main manuscript. Please find the point-to-point responses to your comments below.

Responses to Reviewer #1:

Comment 1: In this manuscript, Corazzi et al. identify an association between somatic CNVs and recurrent DNA break clusters in aphidicolin-treated murine neural progenitor cells. They then go on to show the involvement of DNA polymerase theta in the formation of recurrent DNA break clusters. This was an interesting and well-presented manuscript that was enjoyable to read. My comments for the authors are below.

Response 1: We appreciate their thoughtful review and kind remarks regarding the clarity and significance of our study. We appreciate their thoughtful reading and kind remarks highlighting the clarity and interest of our study. Below, we provide detailed responses to each of the specific comments raised.

Comment 2: Points that I request the authors address and respond to:

I understand the authors' choice of aphidicolin to induce DNA replication stress, as there is very good historical precedent for doing so (in, for example, the detection of common fragile sites). However, it strikes me that the inhibition of DNA polymerase alpha induces a type of replication stress that may not mirror the type of replication stress that could result in CNVs in neural progenitor cells in the absence of any agent. This is particularly pertinent given that replication stress induced by hydroxyurea and aphidicolin generate distinct patterns of somatic CNVs (<https://doi.org/10.1186/s13059-022-02781-0>). I am certainly not suggesting that the authors repeat the analysis with different agents that induce replication stress; that would be quite unreasonable. However, this would be an interesting point to discuss, and it would be helpful to qualify the conclusions appropriately.

Response 2: We thank the reviewer for this insightful comment. We agree that aphidicolin-induced replication stress, resulting from inhibition of DNA polymerase α , may not fully recapitulate the endogenous sources of replication stress occurring in neural progenitor cells under physiological conditions. We appreciate the reference to the recent study highlighting that different agents such as hydroxyurea and aphidicolin can generate distinct CNV landscapes (DOI: 10.1186/s13059-022-02781-0). We included this literature in the revised manuscript on page 2, line 51 (as reference 10):

“Although these DSBs have been proposed as a source of CNVs⁷ with distinct forms of replication stress producing divergent CNV patterns¹⁰, the factors that distinguish DSBs leading to non-recurrent versus recurrent CNVs remain unclear”

We also revised the Discussion to explicitly acknowledge this limitation and to clarify that while aphidicolin provides a well-established experimental model for inducing replication stress and mapping recurrent DNA break clusters, the precise spectrum and mechanisms of CNV formation *in vivo* may differ depending on the underlying cause of replication stress.

Specifically, we have added the following to “Limitation of the study” on page 13, line 482:

*“While aphidicolin is a well-established reagent for inducing replication stress and has been widely used to map common fragile sites, the inhibition of DNA polymerase α may not fully reflect the endogenous forms of replication stress that arise in neural progenitor cells. Distinct replication stressors, such as hydroxyurea, have been shown to produce different CNV patterns (doi: 10.1186/s13059-022-02781-0). Accordingly, our findings should be interpreted as modeling one specific form of replication stress, rather than the complete spectrum that may occur *in vivo*.”*

Comment 3: Figure 1B shows the rank-ordered $\log_2(\text{Aph}/\text{DMSO})$ coverage for every RDC for each of two replicates. While the shape of the curve shows good agreement between replicates, it is not clear to me from this plot whether the rank order itself is well-preserved across replicates. Figure 1C suggests the agreement is generally good, but this would still be nice to show.

Response 3: This is a good point. We have now quantified the ordering. Here, we calculated Pearson’s correlation coefficient between the gene-level paired $\log_2(\text{Aph}/\text{DMSO})$ coverage values of replicate 1 and replicate 2. The correlation between replicates was $r = 0.93$. This significance of this correlation in rankings was ascertained using a permutation test, giving a significance level of $p < 1 \times 10^{-6}$ using 10^6 permutations (the minimal p-value using this number of permutations).

The following has now been added to the manuscript, under the first result section, on page 3, line 112:

“ Closer inspection of the most strongly affected genes showed striking concordance between replicates (Figure 1C). When genes were paired between replicates, correlation in loss of read-depth coverage was very high (Pearson’s $r=0.93$, $p=1 \times 10^{-6}$ permutation tested).”

Comment 4: At the opening of the discussion, the authors state, “In this study, we establish that transcription-dependent RDCs generated under replication stress directly cause somatic CNVs at ultra-long, late-replicating neuronal genes.” This feels a bit strong given the numbers involved: Of the nine CNVs called by Delly2, six of them mapped to an RDC with two not being reproducible across replicates. However, I think there are some fairly straightforward analyses that would help the authors strengthen their conclusions. For instance, what is the probability that the size of CNVs observed overlaps an RDC by random chance? Likewise, it would help if the authors could state how many genes they consider “ultra-long, late-replicating neuronal genes” so that we can get a feel for how many of them show this concordance between CNVs and RDCs.

Response 4: We thank the reviewer for raising this point, which was also mentioned by Reviewer 3. We appreciate the opportunity to clarify this aspect of the analysis.

It may not have been clear in the manuscript that CNV calling with *Delly2* was performed genome-wide, not limited to genes containing RDCs. Consequently, the majority of the genome was tested, yet all CNV events identified were confined to RDC-containing genes. To assess the likelihood of this occurring by chance, we performed a statistical test as suggested by the reviewer.

Specifically, we used a **hypergeometric model**, which, unlike the binomial model that assumes sampling with replacement, quantifies the probability of obtaining n or more successes when sampling without replacement. Across approximately 35,771 genes analyzed, *Delly2* identified six CNVs in genomic regions, which were present in both replicates, all six of which were located within the 152 RDC regions. Under the null hypothesis that CNV formation is independent of RDC regions, the probability of this occurring by chance is $p = 5.33 \times 10^{-15}$.

This extremely low probability supports a strong enrichment of CNV events within RDC-containing genes, consistent with a causal relationship between transcription-dependent RDCs and CNV formation under replication stress.

We have now added the following text to the manuscript, under the first result section, on page 3, line 106:

“Under a null model in which CNV formation is independent of RDC location, the probability that all six recurrent CNVs would lie within RDCs by chance is $p = 5.33 \times 10^{-15}$ (hypergeometric test; population $\approx 35,771$ genes; successes = 152). Thus, while recurrent CNV hotspots are rare at a genome-wide scale, they are strongly enriched at RDC-containing, ultra-long, late-replicating neuronal genes, reinforcing the conclusion that transcription-dependent RDCs provide a proximate source of CNV formation under aphidicolin-induced replication stress.”

We also added a “genome-wide CNV calling” label above revised Figure 1D to clarify CNV analysis scope.

Comment 5: In Figure 5C-D, it is important to see exact p-values to evaluate the results appropriately. If I understand the figure legend correctly, those genes that are significant are hotspots, but based on the other genes and the shape of the distributions, it seems as though these p-values might be quite close to 0.05.

Response 5: The P values for *Lsamp* and *Grid2* in the revised Figure 7D (Figure 5C in the original manuscript) are 0.035 and 0.035, respectively. The P value for DSB density changes in the revised Figure 7E (Figure 5D in the original manuscript) is 0.032. These values have been added to the corresponding figure legend.

Comment 6:

Minor points that should be straightforward to address:

- Line 90-91: A fraction of RDC-containing genes *are* copy number variation hotspots in neural progenitor cells
- Line 652: false *discovery* rate (FDR) correction

Response 6: The first issue has been addressed by replacing it with revised text in the updated manuscript, and the error in the legend of the revised version of Figure 6 has been corrected.

Comment 7: Some readers may find the use of “constant-timing region” (CTR) confusing. First, clearly defining what a CTR is in the main text would make it more accessible to nonspecialists. Second, there is a risk of confusion in the legend of Figure 2 (Lines 615-616) where the grammar suggests that a CTR is a region whose timing is unchanged by aphidicolin treatment, but this is not actually what a CTR is nor is it what the authors mean.

Response 7: We have expanded the description on Page 4, line 130:

“Neural progenitor cells exposed to aphidicolin revealed discrete copy-number losses in six RDC-containing loci, situated within late Constant-Timing Regions (CTRs), where multiple replication units simultaneously complete DNA replication within a single region”

And on Page 22, line 830, under the legend for Figure 2:

“illustrating that each CNV lies within a late constant-timing region (CTR), whose replication timing remained unchanged upon Aph treatment.”

Comment 8: For me, I found the light and dark blue a bit difficult to distinguish between in Figure 1D.

Response 8: We changed the dark blue to orange for revised Figures 1B-D.

Comment 9: For reproducibility, it is useful to specify the exact versions of any software used wherever possible. E.g., Bowtie2, Delly2, etc.

Response 9: We appreciate the suggestion from reviewer 3 to specify the exact versions of any software. We added the following under the Code Availability section:

“Bowtie 2 2.5.0 used for LAM-HTGTS, Repli-seq, and GRO-seq. Samtools 1.7 was used for LAM-HTGTS, and 1.8 and above was used for Repli-seq and GRO-seq. These pipelines and software dependencies were containerized using Docker, ensuring reproducible and consistent version control across analyses. For CNV calling, Delly 1.2.6 is a publicly available package (<https://github.com/dellytools/delly>).”

Comment 10: Having looked through <https://github.com/brainbreaks/>, it’s not completely obvious to me what was and was not used for this manuscript. It looks like there’s more here than was used, and it would help to make the Code Availability statement more specific. That said, the repositories look very nicely documented which is highly commendable. All of the pipelines here look very readable and usable.

Response 10: We thank the reviewer’s positive view on our Git repository, which we spent a significant effort building. We believe that publishing commendable pipelines is essential to achieve FAIR (findable, accessibility, interoperability, and reusability). In this manuscript, we applied three singularity containers for HTGTS (<https://github.com/brainbreaks/HTGTS>), GRO-seq (<https://github.com/brainbreaks/HTGTS>), and Repli-seq (https://github.com/brainbreaks/HighRes_RepliSeq) analyses. In addition, an automated AlignmentAndQC workflow (<https://github.com/DKFZ-ODCF/AlignmentAndQCWorkflows/>) was applied to WGS analysis. This information is included under the Code Availability section.

Responses to Reviewer #2

Comment 11: In this manuscript Corazzi et al. focus on the role of POL θ in the replication-stress-mediated generation of copy-number variations at recurrent DNA break clusters. The origin of these events and their implications for neurological disease and cancer is an intriguing question in the genome instability field. I have, however, reserves regarding the conceptual contribution of the current manuscript.

Response 11: We thank the reviewer for recognizing the relevance of our study to understanding the origins and implications of replication-stress-induced CNVs. We appreciate their concern regarding the conceptual contribution of our work and have carefully considered their detailed comments below. We addressed each of these points in turn, conducting two additional experiments, at the bulk and the single-cell level, to strengthen the evidence for our claims regarding the role of RDCs in CNV formation. The revised manuscript now presents new concepts, beyond what have been shown in prior publications. In addition, we have expanded the analysis of the existing datasets and added clarifications throughout the manuscript to improve conceptual clarity and better articulate the novelty of our findings. New data, analysis, results and conclusions are described in detail below.

Comment 12: The manuscript shows:

- An association between transcription-dependent RDCs induced by replication stress and CNVs. This is already established by several publications, as cited in the manuscript.

Response 12: The reviewer notes that the manuscript reports an association between CNVs and transcription-dependent, replication stress induced RDCs, and that this relationship has been described previously. We agree that this association is consistent with prior work, and is illustrated in Figures 1 and 2, which largely recapitulate established findings. These results are included to validate our experimental system and analytical approach and to provide a foundation for the novel analyses and conclusions presented in the subsequent sections of the manuscript. However, the following experiments characterize the mechanism of RDC-dependent CNVs in greater detail.

We performed perturbation experiments, showing CNV loss is dependent on DNA breaks at RDCs, but not with enhanced replication timing, as implied by previous publications (revised Figure 4). In line with this, DNA breaks have not been measured in prior studies cited in our manuscript, thus our findings do not simply replicate published observations. To highlight this conceptual advance, we have added clarifying text to the Discussion section (Page 11, line 393):

“This study provides experimental evidence that transcription-dependent RDCs induced by replication stress directly generate CNVs. By selectively suppressing transcription at RDC loci, we disrupted RDC formation without altering replication timing, thereby excluding the previously proposed mechanism linking transcription to CNV suppression. Our findings establish a causal relationship, rather than a correlation, demonstrating that a specific subset of CNVs originate from replication stress-mediated DNA breaks triggered by transcription. This work advances current understanding of how genome instability at actively transcribed loci contributes to structural variation.”

Additional response to comment 12:

Although we respectfully disagree with the reviewer's assessment, we reflected on the concern that some readers may share a similar impression. To enhance the impact of this manuscript, **we extended our investigation to determine whether RDC-mediated structural variants penetrate to daughter-cell genomes, and if RDCs cause positionally non-recurrent CNVs, thus contributing to genome diversification.** To this end, we applied a single-cell genomic approach, called Strand-seq, to confirm the inheritance of recurrent CNVs detected in bulk, while additionally revealing non-recurrent, cell-specific structural variants that are not captured by population-level analyses but may nonetheless have biological consequences.

This additional experiment was conducted in collaboration with Jan Korbel's laboratory at EMBL, Heidelberg. Single neural progenitor cells that had undergone one round of DNA replication under aphidicolin or DMSO treatment were subjected to Strand-seq. Strand-seq is an established method for measuring copy number variations at single cell genomes. Single-cell genomics enables the measurement of features that bulk WGS cannot capture, including (1) SV frequency per locus in individual cells, (2) allelic prevalence, and (3) whether DNA copy number changes represent stable structural variants in the genome. We obtained 79 high-quality single-cell libraries from untreated and 88 from aphidicolin-treated XRCC4/p53-deficient cells and used for copy-number analysis (**Figure R1A**, this document).

Figure R1. Detecting structure variants in post-stressed neural progenitor cells at the post-stress cell cycles with Strand-seq.

(A) Schematic illustrating the experimental workflow for single-cell Strand-seq analysis. Neural progenitor cells (NPCs) were treated with aphidicolin (APH) for 96 hours, allowed to recover for 96 hours, and then labeled with BrdU during one round of DNA replication prior to sorting and Strand-seq library preparation to distinguish parental Watson and Crick template strands. **(B)** Dot plot showing gene-level copy number (CN) values for specific RDC-associated genes in individual cells treated with Aph (blue) or DMSO vehicle control (orange). Lower CN values indicate losses at these loci, which are frequent in Aph-treated daughter cells. **(C)** Representative single-cell Strand-seq coverage tracks at the *Lrp1b* locus (chr2) in Aph-treated cells. Top track shows bulk DSB density. Shaded region indicates the *Lrp1b* gene body, displaying focal, haplotype-specific deletions (drops in read coverage on either Watson or Crick strands) that align with DSB clusters. *These figures are presented in the revised manuscript as Figure 3A-C, respectively.*

We first tested whether the six CNV hotspots identified in bulk sequencing persist as structural variants in daughter cells. Using Strand-seq libraries with interpretable Watson/Crick strand inheritance, we detected only a single copy-number loss at *Magi2* in untreated cells, whereas

aphidicolin-treated cells frequently exhibited focal deletions at five of the six bulk-defined hotspots. Multiple independent daughter cells carried distinct intragenic deletion events at these loci—for example, *Lrp1b* losses in 10 cells, *Grid2* in 3 cells, *Magi2* in 4 cells, and single-cell losses at *Cadm2* and *Nrg3*. These events consistently affected only one inherited strand, indicating deletion-type repair of RDC-associated breaks rather than complete locus loss. Statistical testing confirmed that deletion frequencies at these hotspots were significantly elevated relative to background (**Figure R1 B,C**, this document). Thus, copy number variation in these RDC loci were recurrent and not rare.

To assess whether RDC-associated breaks coincide with non-recurrent copy number alteration in single cells, we performed an unbiased CNV analysis on cells with high-quality Watson/Crick strand signals. Cells previously exposed to aphidicolin frequently exhibited substantial DNA loss across the genome (**Figure R2A**, this document), consistent with large, multi-megabase deletions (**Figure R2 B-C**, this document). We therefore focused on cells with pronounced copy-number loss and mapped their large-scale genomic alterations. In 29 cells with reduced genome-wide copy number (mean < 1.92), 302 breakpoints were identified (Supplementary Data 2). A significant fraction of these were mapped to RDC regions (118/302, permutation test, $p=1 \times 10^{-4}$). Notably, nearly half of these RDC-associated breakpoints overlapped with replication timing transition regions (TTRs) (51/118; permutation test, $p = 6 \times 10^{-4}$), indicating that TTR-associated RDCs contribute to substantial copy-number variants.

Figure R2. Arm-level, very large, non-recurrent deletions contain breakpoints significantly associated with RDCs. (A) Scatter plot of mean genome-wide CN in different conditions. Aph-treated cells (blue) exhibit a significant reduction in overall DNA content compared to DMSO controls (orange). (B) Strand-seq plots illustrating large-scale chromosomal terminal deletions (TelDel) whose breakpoints directly coincide with RDC sites (marked by DSB density peaks). Examples shown for *Grik2* (chr10) and *Immp2l* (chr12). Cartoons on the right depict the resulting chromosomal structures. (C) Representative Strand-seq tracks displaying various structural alterations in single cells. Panels are organized as described for (B). *These figures are presented in the revised manuscript as Figure 3D, E, and Supplementary Figure 3B, respectively.*

We also captured two cells that likely came from the same parental cell (Cell 9 and 10), as we observed mirrored Watson/Crick signals on multiple chromosomes, including chromosome 12, showing reciprocal sister chromatid exchange (**Figure R3A**, this document). In one of the suspected daughters (Cell 9), we found only a short Watson peak on chromosome 2 while the Crick strand displayed significant copy number loss downstream of the Watson peak (**Figure R3B**, this document). This feature, as previously seen in hematopoietic stem cells and epithelial cell lines, represented a fold-back chromosome undergoing a break-fusion-bridge cycle. This event was not present on the other daughter cell (Cell 10), suggesting parental cell exposure to replication stress diversified the daughter cell's genomes (**Figure R3C**, this document).

Taken together, these analyses demonstrate that RDC-associated breaks are not restricted to producing focal intragenic deletions, but can also precipitate complex and larger structural variants that extend across megabase-scale chromosomal regions. Rather than being fully repaired or averaged out in a population, lesions initiated at recurrent break clusters are converted into a spectrum of structural outcomes, from small intragenic deletions to arm-scale losses, which segregate into daughter cells. Thus, replication stress at RDCs acts as a driver of genome diversification,

generating both locus-specific and large-scale structural variants that permanently remodel the genomes of neural progenitor cells.

We have included these new results in the revised manuscript, under the result section “**Large single-cell structure variants co-localize with RDCs**”, Figure 3 and supplementary Figure 3, as well as included the methods. Additional co-authors (Eva Benito, Marco Raffaele Cosenza, Patrick Hasenfeld, Thomas Weber, Jan O. Korbel) who contributed to Strand-seq experiments and analyses were added to the revised manuscript.

Comment 13: The involvement of POL θ in generating RDC-associated translocations in NHEJ-deficient background. It is well-known that POL θ -mediated repair (by MMEJ) acts as a backup for NHEJ. The contribution of showing that this also occurs in this particular context is minor.

Response 13: We agree that theta-mediated end-join (TMEJ) is not the main focus of this manuscript. Our aim was not to re-establish the pathway hierarchy between NHEJ and MMEJ, but rather to define how RDCs contribute to genome instability and diversification, with mechanistic insight into how these breaks are resolved. To make the findings in TMEJ more accessible to the readers, we revised the manuscript with the additional text under discussion, starting from page 12, line 428:

“In these contexts, MMEJ activity at a genome-wide scale across replication time zones in response to endogenous, cell-intrinsic DSBs were not defined. Our findings expand this framework by demonstrating a broad contribution of Pol θ -mediated repair function in TTR and CTR (Fig. 6C). Following this notion, we demonstrated that the majority of translocations in XRCC4/p53-deficient cells rely on 2 base-pair microhomology usage (Fig. 6F); inhibiting the polymerase activity of Pol θ diminished the microhomology usage, converting it to the scale of NHEJ. Our results align closely with those of a very recent study focused on an immunoglobulin gene locus in the B cells genome (DOI: 10.1038/s41467-025-65555-9), consolidating that Pol θ contributes to DSB repair across cell cycle phases.

Although biochemical analysis suggested that Pol θ stabilizes the DSB ends (doi:10.1093/nar/gkac1201), , this effect has not been measured in cycling cells. We observed a ~20-base shortening of bait-derived DNA fragments (Fig. 6H), closely aligning with the biochemical properties characterized in vitro (doi:10.1016/j.molcel.2022.09.013; doi:10.7554/eLife.13740). These findings suggest that Pol θ may be required either to convert single-stranded DNA ends into double-strand breaks, or that ART558-mediated inhibition disrupts Pol θ -driven microhomology stabilization. In addition, the end protection ability of Pol θ is independent from NHEJ status, as the same trend was seen in the wild-type neural stem/progenitor cells (Fig. 7G).”

Comment 14: The involvement of POL θ in preventing RDC-associated translocations in wild-type background. This is the most interesting observation, which the authors explain by a potential role of POL θ in preventing the emergence of replication-associated DSBs. Although this is not completely novel, as the authors indicate, it would be worth exploring further.

Response 14: We thank the reviewer for their encouragement in exploring the role of Pol θ in preventing the emergence of replication-associated damage. To investigate whether Pol θ plays an NHEJ context-dependent role in recurrent CNV formation, we first performed whole-genome sequencing on Xrcc4/p53-deficient neural progenitor cells treated with a Pol θ inhibitor alone, or in combination with aphidicolin. In this NHEJ-deficient background, all six recurrent CNVs induced by

aphidicolin were completely abolished upon Pol θ inhibition, demonstrating that Pol θ activity is required for replication stress–induced CNV formation in the absence of NHEJ (**Figure R4A**, this document).

Next, we performed whole-genome sequencing on wild-type neural stem/progenitor cells treated with (1) solvent control (DMSO), (2) aphidicolin alone, (3) Pol θ inhibitor alone, or (4) aphidicolin plus Pol θ inhibitor. Aphidicolin treatment resulted in five reproducible CNVs in wild-type cells. In contrast to the NHEJ-deficient context, Pol θ inhibition neither reduced nor enhanced recurrent CNV formation in wild-type cells and did not induce additional CNVs (**Figure R4B**, this document). Together, these results indicate that Pol θ mediates replication stress–induced CNV formation specifically when NHEJ is compromised, whereas intact NHEJ suppresses Pol θ –dependent CNV generation.

In addition, Pol θ inhibition alone resulted in a strong DNA copy number alteration phenotype that is associated with replication stalling. In this context, DNA copy number gain and loss became inextricably linked to DNA replication timing - CNV gains are located at early-replicating regions, while CNV losses appear to be late-replicating (**Figure R5A,B**, this document). Strikingly, Pol θ inhibition alone in NHEJ-proficient cells caused more severe DNA content distortions than in NHEJ-deficient cells (**Figure R5C**, this document), suggesting that NHEJ constrains Pol θ –dependent replication progression under replication stress.

We propose that sustained Pol θ inhibition activates NHEJ-mediated ligation of DNA breaks, a pathway previously shown to operate at stalled replication forks, restoring DNA continuity but leaving nascent single-stranded gaps unresolved, thereby preventing replication restart and effectively freezing replication intermediates. In contrast, in the absence of NHEJ, unrepaired breaks persist at stalled forks, allowing exposed single-stranded DNA, particularly on the leading strand, to engage recombination-based restart pathways, including single-strand annealing between sister chromatids. Consistent with this model, combined loss of Lig4 and Pol θ has been shown to enhance single-strand annealing–mediated repair events (DOI:10.1038/ncomms16112), supporting a pathway switch from end joining to homologous mechanisms when both Pol θ and NHEJ are compromised. In addition, we speculated that persistent aphidicolin exposure generates extensive nascent single-stranded DNA gaps that exceed the repair capacity of Pol θ , thereby forcing a switch to Pol θ –independent fork restart or remodeling pathways.

We have included the above-mentioned findings to the revised manuscript under the main text (corresponding to revised Figures 5 and 7), and in the corresponding discussion sections (from page 12, between lines 462 - 479).

Comment 15: Therefore, in my view, the manuscript is confirmatory and preliminary, having made an interesting observation that would need to be studied in more detail to provide novel insights that can constitute a true advancement to our mechanistic knowledge on the sources of RCD-associated CNV.

Response 15: We believe that this concern has been resolved with new data presented in the revised manuscript. With new evidence from Strand-seq and whole-genome sequencing, this study demonstrates that RDCs promote copy number losses in an NHEJ and Pol θ context-dependent manner, and creates genome diversity. We have revised the manuscript accordingly across every section to make sure these notions became accessible for readers. These changes, including those mentioned above, are highlighted in blue in the revised manuscript.

Concretely, our work provides four main advances:

1. From average copy number losses in bulk to single-cell precision:

We provided single-cell–resolved evidence that replication stress at recurrent DNA break clusters (RDCs) is not merely a marker of genome fragility but a direct driver of heritable genome diversification. Using Strand-seq to trace structural outcomes across daughter cells, we show that RDC-associated lesions generated in one cell cycle are fixed as stable deletion-type structural variants in the next, segregating asymmetrically between daughters. These outcomes span a continuum from focal intragenic deletions at known CNV hotspots to large, terminal, and arm-scale chromosomal losses, with breakpoints highly enriched at RDCs. Importantly, many of these heterogeneous large-scale events escape detection by bulk sequencing. We further reveal that RDC-initiated breaks can seed ongoing genome instability via break–fusion–bridge cycles, linking transient replication stress to sustained structural evolution. Together, our findings redefine RDCs as active engines of somatic genome diversification in neural progenitor cells, rather than passive correlates of replication stress.

2. From correlation to causation:

We directly link transcription-dependent RDCs to CNV formation in the *same* neural progenitor cell system, showing that (i) all reproducible aphidicolin-induced CNVs lie within pre-defined RDCs, and (ii) CRISPR-mediated promoter/enhancer deletions that abolish transcription and RDC formation at *Ctnna2* and *Nrg3* also eliminate the corresponding CNVs, without advancing

replication timing. This establishes RDCs as necessary intermediates in replication-stress-induced CNV formation at these loci and clarifies that late replication timing per se is not sufficient.

3. Mechanistic dissection the role of Pol θ in RDC-dependent CNV formation:

We show that Pol θ 's polymerase activity is required to maintain RDC-derived end accessibility and microhomology-mediated joining in XRCC4/p53-deficient neural progenitor cells, and that, in contrast, Pol θ suppresses RDC-derived DSB capture in wild-type neural stem/progenitor cells. This dual role integrates Pol θ 's established function in alternative end joining with emerging evidence for its involvement in replication fork protection, specifically in the context of transcription-replication conflicts at long neural genes.

4. Refined model for how replication stress generates recurrent and non-recurrent CNVs at neural RDC genes:

By combining high-coverage WGS, LAM-HTGTS, GRO-seq, Repli-Seq, and genetic perturbations of transcription and Pol θ , we propose a model in which transcription-driven RDCs at long, late-replicating, LAD-associated genes are the principal sources of replication-stress-induced CNVs in neural progenitors, and in which Pol θ modulates both break formation and end processing in a context-dependent manner (Figure 7H).

Reviewer #3

Comment 16: To investigate the relationship between replication stress-induced DSBs and CNVs, Corazzi et al. employed ultra-deep whole-genome sequencing to identify CNV hotspots upon replication inhibitor treatment in neural stem and progenitor cells. Building on their previously defined recurrent DNA break clusters (RDCs), they observed that a subset overlapped with Aph-induced CNV hotspots and that these RDC/CNV hotspots were transcription-dependent. They further examined the contribution of Pol θ , an MMEJ factor, to CNV generation in both NHEJ-deficient and WT contexts. Collectively, they conclude that RDCs may represent a potential source of CNVs.

My main concern, however, is the overall prevalence of these RDC-induced CNVs. Are they relatively common or rather rare events? Specifically, the manuscript does not provide the total number of genome-wide CNV hotspots; instead, the analysis is restricted to RDC-related CNVs. It would be important to report the proportion of RDC-associated CNVs relative to all CNVs detected. Moreover, in their previous study (Nat. Commun. 2024), the authors identified 152 RDCs, most located at TTRs, CTRs, and IZs, which generally do not overlap with CNV hotspots. Only 5 out of 152 RDC loci were reported to exhibit CNV formation. This raises the question of whether the contribution of RDCs to CNV generation is quantitatively minor, and further clarification and contextualization would strengthen the manuscript.

Response 16: We thank the reviewer for raising this important point regarding the overall prevalence of RDC-associated CNVs. This same issue was also independently noted by Reviewer 1, and we fully appreciate that the manuscript was not sufficiently clear on this aspect:

CNV calling with Delly2 was performed genome-wide, not biased or restricted to RDC-containing genes. Across approximately 35,771 genes, Delly2 identified six recurrent CNV hotspots that were present in both independent Aph-treated replicates. All six of these hotspots were located within the previously defined set of 152 RDC loci. Thus, 100% (6/6) of genome-wide recurrent CNV hotspots

overlapped RDC regions, while 3.9% (6/152) of RDCs showed recurrent CNVs under these conditions.

To assess the likelihood that this enrichment could arise by chance, we used a hypergeometric model to quantify the probability of observing n or more CNV hotspots within RDC loci when drawing six hotspots from the genome without replacement. Under the null hypothesis that CNV formation is independent of RDC location, the probability that all six CNV hotspots would fall within the 152 RDC loci by chance is $p = 5.33 \times 10^{-15}$. Although recurrent CNV hotspots are therefore rare at the genome-wide level, they are extremely strongly enriched within RDC regions, consistent with a mechanistic relationship between transcription-dependent RDCs and CNV formation during replication stress.

We are grateful to the reviewers for prompting us to make this relationship clearer, and we have now expanded the Results section to explicitly state the single-cell and genome-wide nature of these analyses, the corresponding proportions, and the statistical test used. Specifically, we added the text below under the result section, on page 3, line 106:

“Under a null model in which CNV formation is independent of RDC location, the probability that all six recurrent CNVs would lie within RDCs by chance is $p = 5.33 \times 10^{-15}$ (hypergeometric test; population $\approx 35,771$ genes; successes = 152). Thus, while recurrent CNV hotspots are rare at a genome-wide scale, they are strongly enriched at RDC-containing, ultra-long, late-replicating neuronal genes, reinforcing the conclusion that transcription-dependent RDCs provide a proximate source of CNV formation under aphidicolin-induced replication stress.”

We also conducted new single-cell genomics experiments to assess the penetration of RDC-CNV to daughter cells, and whether there are other frequent, positionally non-recurrent structural variants associated with RDCs. To this end, we applied Strand-seq to identify structural variants in individual cells post aphidicolin treatment. A detailed description of the experiment set up can be found under **Comment 12 (pages 5-9, this document)**, the response to reviewer 2, as well as in the revised Result section related to revised Figure 3.

In summary, in 88 post aphidicolin-treated cells, we found *Lrp1b* losses in 10 cells, *Grid2* in 3 cells, *Magi2* in 4 cells, and single-cell losses at *Cadm2* and *Nrg3*. This finding indicated that CNVs are recurrent in these genes. In addition to CNVs at late CTR, we also found large-scale, frequent while positionally non-recurrent copy number variations involving breakpoints in TTR-containing RDCs (**Figures R1-3** this document, between pages 6-8). Given that over half of cells harboring copy-number–altering structural variants show breakpoint enrichment at RDCs, RDC-origin CNVs can be considered abundant at single cell resolution. We have included these new results in the revised manuscript, under section called **“Large single-cell structure variants co-localize with RDCs”** starting from Page 5, line 159.

Comment 17:

Other comments:

1. The WGS experiments were performed in cells treated with 0.5 μ M aphidicolin (Aph) for 72 h followed by 0.25 μ M Aph for 24 h. The rationale for changing the dose after 72 h is unclear. Could this treatment regimen primarily generate under-replicated regions rather than bona fide CNVs? Performing WGS in G1-arrested cells after Aph treatment would help address this concern.

Response 17: We apologise for the confusion regarding the Aph treatment concentration. The actual concentration used for the WGS is 0.5 μ M for 96 hours as described in the material and method. Therefore, we have revised the following accordingly:

Figure 1A, Page 22, lines 805:

“Experimental design. Mouse ESC-derived NPCs were treated with 0.5 μ M aphidicolin (Aph) for 96 h.”

Comment 18: Some of the interpretations appear overstated. For example, the conclusion that Pol θ suppresses RDC-prone hotspots is not supported by direct evidence. Similarly, they showed the CNVs/RDC are transcription dependent at two sites, then conclude “DNA sequence loss in CNV hotspots is RDC-dependent”, which could be correlative than causal.

Response 18: We have revised the text to avoid overstating Pol θ suppresses RDC-prone hotspots across the manuscript. In addition, **we conducted new WGS experiments** to investigate whether Pol θ plays an NHEJ context-dependent role in recurrent CNV formation. We found that in the NHEJ-deficient background, all six recurrent CNVs induced by aphidicolin were completely abolished upon Pol θ inhibition, demonstrating that Pol θ activity is required for replication stress-induced CNV formation in the absence of NHEJ (**Figure R4A**, this document). Whereas in the wild-type neural stem/progenitor cells, inhibiting Pol θ activity did not affect CNV formation (**Figure R4B**, this document).

These findings are summarized under the revised result section, under page 7, line 238:

“Pol θ promotes CNV in XRCC4/P53-deficient neural progenitors

Next, we sought to determine whether polymerase theta (Pol θ) activity regulates CNV formation in an NHEJ-deficient context. To this end, we treated XRCC4/p53-deficient neural progenitor cells with aphidicolin alone or with aphidicolin plus ART558 (DOI:10.1038/s41467-021-23463-8), a small-molecule inhibitor of Pol θ DNA synthesis activity. Combined treatment with aphidicolin and ART558 abolished focal, recurrent CNV formation at six aphidicolin-dependent CNV loci (Fig. 5), suggesting aphidicolin-mediated CNVs were Pol θ -dependent in XRCC4/p53-deficient cells. Unbiased, genome-wide copy-number profiling showed that a subclonal chromosome 8 gain detected in untreated cells was lost, leading to reduced \log_2 coverage ratios (Supplementary Data 1; Supplementary Fig. 5A). We also detected a reproducible subclonal loss of chromosome 6, consistent with selection in culture. Further, we identified 46 additional reproducible CNVs (Supplementary Data 1). These CNVs were absent in cells treated with aphidicolin alone but were consistently detected in cells treated with ART558, indicating that they arise specifically upon Pol θ inhibition. In addition, more than half (20/46) of the Aph/ART558-induced CNVs were not in genes, suggesting they arise from an RDC-independent mechanism.

Because Pol θ is known to fill nascent DNA gaps at stalled replication forks, particularly on the lagging strand (DOI:10.7554/elife.13740, DOI:10.1016/j.molcel.2022.09.013), we hypothesized that these copy-number alterations reflect differences in DNA content resulting from replication stalling. To test this, we examined replication timing at recurrent CNV loci in cells treated with both aphidicolin and ART558. We observed a significant correlation in which CNV gains were associated with early-replicating regions (Kendall’s rank correlation, tau = 0.31, p = 0.002), whereas CNV losses were enriched in late-replicating regions, indicating that copy-number changes mirror replication timing (Supplementary Fig. 5B). This relationship was enhanced in cells treated with

ART558 alone (Kendall's rank correlation, $\tau = 0.51$, $p < 2.2 \times 10^{-16}$; Supplementary Fig. 5C), supporting the notion that ART558-induced CNVs were direct consequences of replication stalling."

And page 9, line 314:

*"In order to investigate whether the role of Pol θ in recurrent CNV formation is NHEJ-deficiency dependent, we performed the same whole genome sequencing experiments for wild-type neural stem and progenitor cells isolated from embryonic mouse brains (Fig. 7A). We identified four recurrent CNVs (Fig. 7A) in aphidicolin-treated cells, three (*Lsamp*, *Lrp1*, and *Csmd1*) of which overlapped with RDCs identified in *XRCC4/p53*-deficient neural progenitor cells. We also found that the *Grid2* locus expressed a significant CNV in one of the two repeats. In contrast to *XRCC4/p53*-deficient cells, combinatory treatment of aphidicolin and Pol θ inhibition had no effect on recurrent CNV formation (Fig. 7A), nor did the treatment induce new CNVs. By contrast, Pol θ inhibition alone caused a severe replication-stalling phenotype in which DNA copy-number profiles significantly tracked replication timing (Kendall's rank correlation, $\tau = 0.51$, $p < 2.2 \times 10^{-16}$; Supplementary Fig. 7A,B), exceeding the magnitude observed in NHEJ-deficient cells. These results imply that the NHEJ pathway inhibits Theta-dependent fork progression."*

Comment 19: No data directly demonstrate the effect of Pol θ inhibition on CNV formation. The influence of Pol θ at RDCs is described as general, without distinguishing features specific to CNV-associated RDCs.

Response 19: We performed whole genome-sequencing assay on *Xrcc4/p53*-deficient neural progenitor cells as well as in the wild-type neural stem/progenitor cells. A similar point was raised by reviewer 2, as such results addressing this issue described above, under **Comment 14** and **Figures R4-5** in this document. We found that Pol θ inhibition abolished CNVs at the six CNV hotspots in *XRCC4/p53*-deficient cells, while the inhibition had no effect on CNV formation in the wild-type cells. These results were included in the revised manuscript, under Figures 5A, 7A, and supplementary Figures 5A and 7A.

Comment 20: The manuscript does not address why only a subset of RDCs give rise to CNVs while the majority do not. Clarification or speculation on this discrepancy would be valuable.

Response 20: To assess whether RDC-associated breaks coincide with non-recurrent structural variants associated with other RDCs, particularly RDCs outside of the six late-replicating CNV regions, we performed an unbiased CNV analysis on single cells derived from Strand-seq experiments (**Figure R1**, this document). Strand-seq is an established method for measuring copy number variations at single cell genomes. Single-cell genomics enables the measurement of features that bulk WGS cannot capture, including (1) SV frequency per locus in individual cells, (2) allelic prevalence, and (3) whether DNA copy number changes represent stable structural variants in the genome.

We obtained 79 high-quality single-cell libraries from untreated and 88 from aphidicolin-treated *XRCC4/p53*-deficient cells and used for copy-number analysis (**Figure R1A**, this document). We confirmed that the six CNV hotspots identified in bulk sequencing persist as structural variants in daughter cells. (**Figure R1 B,C**, this document). We also confirmed that the focal, recurrent structural variants occur in one third of the daughter cells, indicating that copy number variation in these RDC loci were recurrent and not rare.

In addition, around one third of aphidicolin treated cells showed substantial DNA loss across the genome (**Figure R2**, this document). This pattern is consistent with arm-level, or at least multi-megabase, deletions. We therefore focused on cells with pronounced copy-number loss, and mapped their large-scale genomic alterations. 29 cells, including 2 cells in the untreated condition, had a mean copy number across the genome less than 1.92. We found that, of 302 breakpoints found in these cells, 118 were located in RDC regions, representing a highly significant enrichment (permutation test, $P=1 \times 10^{-4}$). Notably, approximately half of these breakpoints overlapped replication timing transition regions (TTRs) (51/118; permutation test, $p = 5 \times 10^{-4}$), indicating that TTR-associated RDCs contribute to severe copy-number variants.

We also captured two cells that likely came from the same parental cell (Cell 9 and 10), as we observed mirrored Watson/Crick signals on chromosome 12, showing reciprocal sister chromatid exchange (**Figure R3A**, this document). In one of the suspected daughters (Cell 9), we found only a short Watson peak on chromosome 2 while the Crick strand displayed significant copy number loss downstream of the Watson peak (**Figure R3B,C**, this document). This feature, as previously seen in hematopoietic stem cells and epithelial cell lines, represented a fold-back chromosome undergoing a break-fusion-bridge cycle. This event was not present on the other daughter cell (Cell 10), suggesting parental cell exposure to replication stress diversified the daughter cell's genomes.

Taken together, these analyses demonstrate that RDC-associated breaks are not restricted to producing focal intragenic deletions, but can also precipitate complex and larger structural variants that extend across megabase-scale chromosomal regions. Rather than being fully repaired or averaged out in a population, lesions initiated at recurrent break clusters are converted into a spectrum of structural outcomes, from small intragenic deletions to arm-scale losses, which segregate into daughter cells. Thus, replication stress at RDCs acts as a driver of genome diversification, generating both locus-specific and large-scale structural variants that permanently remodel the genomes of neural progenitor cells. We added these new results in the revised manuscript, under the "**Large single-cell structure variants co-localize with RDCs**" section, starting from page 5, line 159.

Comment 21: Certain statements are confusing or imprecise. For instance, the subtitle "RDC accessibility" is unclear, and the phrasing "Six of these deletions mapped directly within an RDC showing significant DNA sequence losses" is ambiguous, does this refer to six distinct RDCs or six deletions within a single RDC?

Response 21: We thank the reviewer for pointing out the issue. We rephrased the sentence on page 3, line 119 to clarify the deletion position:

"Each of these six deletions mapped individually to a distinct RDC-containing gene, showing significant DNA sequence loss (Magi2, Nrg3, Lrp1b, Grid2, Cadm2, and Cttna2) "

We also changed the result subtitle to "*Pol θ promotes RDC-mediated translocation in XRCC4/P53-deficient cells*" on page 7, line 264.

Response to Reviewers (NCOMMS-25-64474B)

Corazzi L. et al., Recurrent DNA Break Clusters Drive Replication-Stress-Induced Copy Number Variants and Genome Diversification.

Remark for all reviewers:

We thank the reviewers for appreciating the additional work, text revisions, and the clarity of the pipeline schematics. We also appreciate the reviewers' support for publishing this work. We will take advantage of future opportunities to explore the remaining mechanistic questions.